# DART: Difficulty-Adaptive Reasoning Truncation for Efficient Large Language Models

## Abstract

Adaptive reasoning is essential for aligning the computational effort of large language models (LLMs) with the intrinsic difficulty of problems. Current chain-of-thought methods boost reasoning ability but indiscriminately generate long explanations, leading to evident inefficiency. However, existing reinforcement learning approaches to adaptive thinking remain unstable and heavily reward-dependent. Here we propose **DART**, a supervised **D**ifficulty-**A**daptive **R**easoning **T**runcation framework that adjusts thinking length according to problem difficulty. By distilling concise reasoning patterns from stronger models, interpolating them into a continuum of reasoning styles, and curating optimal training data that balances correctness and compactness, DART learns when to "stop thinking". Across multiple mathematical benchmarks, experimental results demonstrate its remarkable efficiency while preserving or improving accuracy, achieving a significant 81.2% reasoning truncation (DeepSeek-R1-Distill-Qwen-7B on GSM8K dataset) with $5.33\times$ computational acceleration. DART provides a stable and general paradigm for efficient reasoning, advancing the development of adaptive intelligence in LLMs.

## 1 Introduction

The emergence of chain-of-thought (CoT) reasoning has marked a significant advance in enhancing the problem-solving abilities of LLMs by decomposing complex questions into intermediate steps (Wei et al., 2022; Kojima et al., 2022). Despite its effectiveness, the conventional CoT paradigm exhibits a critical inefficiency: it typically generates reasoning chains of a fixed, often excessive, length regardless of the inherent difficulty of the problem at hand (Chen et al., 2024; Fan et al., 2025; Sui et al., 2025). This "one-size-fits-all" approach results in substantial computational redundancy, increasing inference latency and resource consumption—a major bottleneck for deploying LLMs in applications.

Adaptive reasoning, which aligns computational effort with problem difficulty, offers a promising path toward efficiency. Recent attempts to achieve such adaptability have largely relied on reinforcement learning (RL) frameworks, training models to penalize unnecessary reasoning length while preserving accuracy (Arora & Zanette, 2025; Ling et al., 2025; Zhang et al., 2025b; Shen et al., 2025). However, these RL-based methods remain unstable and heavily reward-dependent, suffering from high training difficulty and limited generalizability. Alternative approaches such as knowledge distillation focus on generating shorter reasoning chains (Yu et al., 2024; Kang et al., 2025; Xia et al., 2025), yet they produce statically compressed CoTs that lack dynamic adaptability. The key challenge is therefore to enable difficulty-aware reasoning in a stable and general manner, without relying on complex RL pipelines.

To address this gap, we propose **DART** (**D**ifficulty-**A**daptive **R**easoning **T**runcation), a supervised learning framework that enables LLMs to dynamically adjust their reasoning length according to problem difficulty. As illustrated in Fig. 1, DART bypasses the instability of RL through a structured data-centric **pipeline**: i) **Reasoning Distillation**: concise reasoning chains are distilled from a powerful teacher model, providing

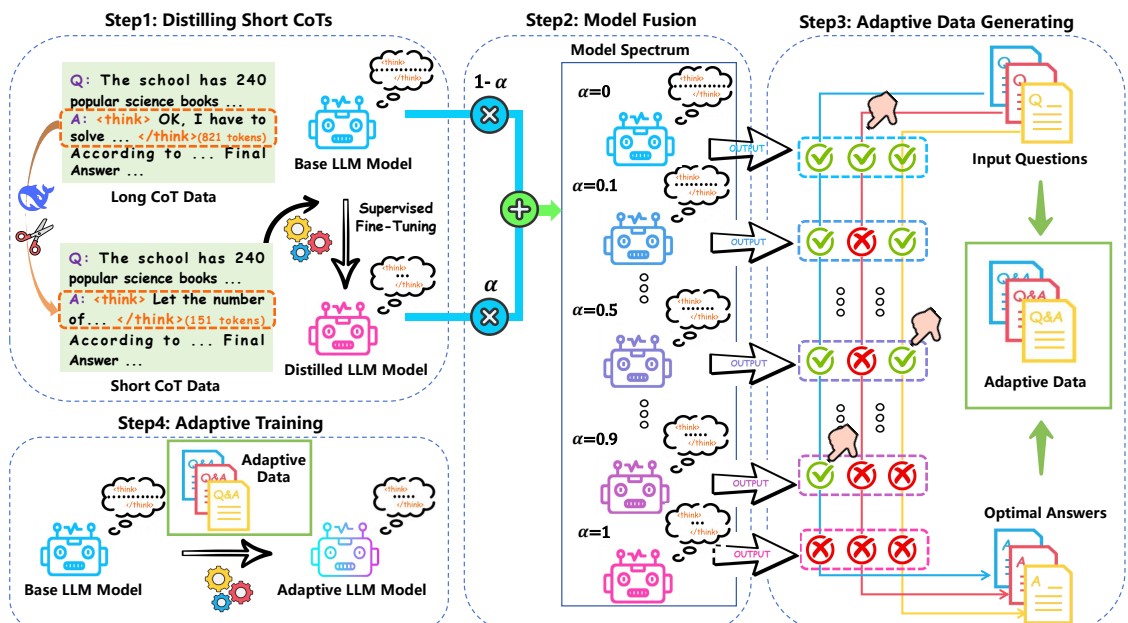

Figure 1: Overall workflow of the proposed DART framework.

a base model that learns compact yet faithful reasoning; ii) **Reasoning Interpolation**: by interpolating between the base and distilled models with a coefficient $\alpha$, we generate a continuum of reasoning styles ranging from verbose to concise; iii) **Optimal Data Curation**: for each problem, the shortest reasoning chain that still yields the correct answer is automatically selected, intrinsically matching depth to difficulty; iv) **Supervised Adaptive Training**: a final model is trained on this curated dataset, learning to "stop thinking" on the minimal sufficient step without reward engineering. Across representative mathematical benchmarks, DART reduces reasoning length by up to 81.2% (DeepSeek-R1-Distill-Qwen-7B on GSM8K dataset) with $5.33\times$ computational acceleration, establishing a stable paradigm for efficient adaptive reasoning in LLMs.

## 2 RELATED WORK

**Reasoning Chain Compression and Distillation.** To address the computational overhead of lengthy reasoning chains, several compression and distillation techniques have been proposed. Knowledge distillation methods train smaller student models to mimic the reasoning processes of larger teacher models (Yu et al., 2024; Kang et al., 2025; Xia et al., 2025). These approaches typically produce statistically compressed reasoning chains that maintain a fixed length across all problems. Prompt-based compression techniques (Xu et al., 2025; Han et al., 2024) attempt to generate shorter reasoning chains through specialized prompting strategies, but often struggle with accuracy preservation on complex problems. Our work differs by introducing dynamic compression that adapts reasoning length to problem difficulty, achieving better efficiency-accuracy trade-offs than static compression methods.

**Adaptive Reasoning Methods.** Adaptive reasoning aims to dynamically adjust the reasoning process based on problem characteristics. Reinforcement learning approaches (Arora & Zanette, 2025; Yu et al., 2025; Shen et al., 2025; Zhang et al., 2025a) train length policies to decide when to halt reasoning, but face challenges with training stability and reward engineering. Prompt-based adaptive methods (Han et al., 2024) use heuristic rules to control reasoning length, but lack learning-based optimization. Unlike these approaches, DART employs a novel supervised learning framework that learns optimal reasoning length policies from automatically curated data, avoiding the instability of RL methods while maintaining architectural flexibility.

# 3 METHOD

## 3.1 MOTIVATION

The core motivation behind our work stems from a fundamental observation on the relationship between the amount of reasoning (measured in the number of tokens generated) and the final reasoning accuracy.

To quantify this, we conducted a preliminary experiment on a subset of 10 problems from the MATH-500 (Lightman et al., 2023) dataset. Using our model fusion technique (detailed in Section 3.3), we generated reasoning chains for these problems across a spectrum of lengths, controlled by the fusion coefficient $\alpha$. For each problem, we evaluated the correctness of the answer yielded by chains of varying lengths. We then analyzed the aggregate accuracy as a function of the average number of tokens in the reasoning chains.

The key observation is that the resulting accuracy-vs-tokens curve exhibits a pronounced Sigmoid (S-shaped) pattern, as illustrated in Fig. 2. This curve reveals three distinct phases: i) an initial low-accuracy plateau with insufficient reasoning steps, ii) a rapid ascent where token increases significantly boost accuracy (the "sweet spot"), and iii) a final plateau of diminishing returns where additional tokens yield minimal gains.

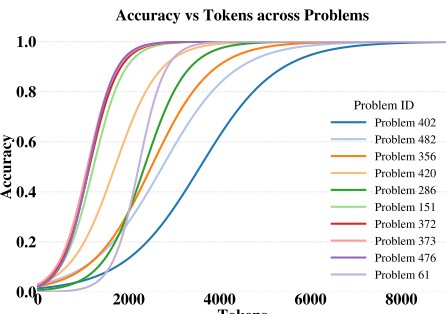

This Sigmoid relationship highlights a critical inefficiency in standard CoT generation: for given problem, there exists an optimal reasoning length that is sufficient to achieve the highest accuracy, and generating anything beyond this point is wasteful.

Figure 2: The trade-off between reasoning chain length and accuracy, illustrated with 10 problems from the MATH-500 dataset.

However, this optimal length is not universal; it is tied to the difficulty of the individual problem. Simple problems may reside on the upper plateau, requiring only a few steps, while complex ones may need to be in the rapid ascent phase to be solved correctly. Current methods—whether generating long chains or distilling fixed short chains—operate at a single point on this curve. They force all problems to be processed with a one-size-fits-all reasoning budget, inevitably leading to inefficiency (for easy problems) or inadequacy (for hard ones).

Therefore, the goal is not to find a single best average length, but to enable a model to dynamically locate the minimal sufficient point on this curve for each input problem. This is the definition of adaptive thinking that our method, DART, is designed to achieve.

## 3.2 STEP 1: DISTILLING SHORT COTS

This initial step aims to establish the two core model components for the DART framework: a compact **Base Model** that possesses inherent long-chain reasoning capability, and a **Distilled Model** derived from it that produces concise reasoning chains. This distillation process is crucial for creating the "short-chain" endpoint of the reasoning spectrum, which will be interpolated with the powerful pre-existing "long-chain" base model.

**Base Model** ($M_{\text{base}}$). We select an efficient, open-source base model that natively supports chain-of-thought reasoning, such as DeepSeek-R1 (Guo et al., 2025) series or Qwen3 (Yang et al., 2025) series. This model, denoted as $M_{\text{base}}$, is employed as-is for its ability to generate detailed, step-by-step reasoning. It serves as the efficient backbone whose behavior we aim to adapt, representing the long-chain reasoning style.

**Short CoT Data Distillation** ($\mathcal{D}_{\text{short}}$). We construct the short-chain dataset by compressing existing high-quality CoT data. Using a powerful distillation teacher model (e.g., DeepSeek-V3 (Guo et al., 2025)), we shorten each long reasoning chain $\text{CoT}_i^{\text{long}}$ to a concise version $\text{CoT}_i^{\text{short}}$ while preserving logical

correctness:

$$\text{CoT}_i^{\text{short}} = M_{\text{distillation}-\text{teacher}}(\text{Prompt}_{\text{compress}}(x_i, y_i, \text{CoT}_i^{\text{long}})) \tag{1}$$

The resulting dataset retains the original question $x_i$ and answers $y_i$ but contains significantly shortened reasoning paths.

**Training the Distilled Model** ($M_{\text{distilled}}$). The Distilled Model ($M_{\text{distilled}}$) is created by performing supervised fine-tuning (SFT) on the original $M_{\text{base}}$ using the compressed dataset $\mathcal{D}_{\text{short}}$. This model learns to produce accurate answers with minimal reasoning, representing the "short-chain" endpoint. The learning objective is to adapt the base model's reasoning behavior to generate concise chains:

$$\mathcal{L}_{\text{distilled}} = -\sum_t \log P(w_t^{\text{short}}|x, w_{<t}^{\text{short}}) \tag{2}$$

By the end of this step, we have two models that share an architectural origin but possess distinct "reasoning styles". $M_{\text{base}}$ embodies thoroughness at the cost of efficiency, while $M_{\text{distilled}}$ embodies extreme efficiency. The interplay between these two models enables the adaptive capabilities developed in the subsequent steps.

## 3.3 STEP 2: CREATING A MODEL SPECTRUM VIA FUSION

Having established two distinct endpoints of the reasoning spectrum—$M_{\text{base}}$ for long, thorough chains and $M_{\text{distilled}}$ for short, concise ones—we now aim to generate reasoning chains of intermediate lengths. To achieve this in a computationally efficient and stable manner, we employ model fusion (Ilharco et al., 2022), a technique that interpolates between the parameters of two models derived from the same pre-trained checkpoint. The core idea is to create a continuum of models, $M_\alpha$, controlled by a fusion coefficient $\alpha \in [0, 1]$, where each fused model exhibits a unique balance between the reasoning styles of $M_{\text{base}}$ and $M_{\text{distilled}}$. This approach is superior to training a separate model for each desired length, as it requires no additional training and only a linear combination of parameters.

**Fusion Process.** For any given $\alpha$, the parameters $\theta_\alpha$ of the fused model $M_\alpha$ are calculated as a weighted linear combination of the parameters of the base and distilled models:

$$\theta_\alpha = (1 - \alpha) \cdot \theta_{\text{base}} + \alpha \cdot \theta_{\text{distilled}} \tag{3}$$

where $\theta_{\text{base}}$ are the parameters of $M_{\text{base}}$, $\theta_{\text{distilled}}$ are the parameters of $M_{\text{distilled}}$, and $\alpha$ is the fusion coefficient. The resulting model $M_\alpha$ is instantiated with the parameters $\theta_\alpha$.

**Interpretation of the Fusion Spectrum.** The coefficient $\alpha$ dictates the "reasoning identity" of the fused model: i) When $\alpha = 0$, $M_{\alpha=0} = M_{\text{base}}$. This model generates the longest, most detailed chains. ii) When $\alpha = 1$, $M_{\alpha=1} = M_{\text{distilled}}$. This model generates the shortest, most compressed chains. iii) When $0 < \alpha < 1$, the model $M_\alpha$ blends the behaviors of its parents. As $\alpha$ increases from 0 to 1, the generated reasoning chains become progressively shorter and more concise, effectively traversing the accuracy-vs-tokens curve introduced in Section 3.1.

## 3.4 STEP 3: CURATING THE ADAPTIVE TRAINING DATA

The model spectrum $M_\alpha$ generated in the previous step provides a diverse set of reasoning strategies for any given problem. The goal of this step is to automate the construction of a high-quality training dataset $\mathcal{D}_{\text{adaptive}}$ where each problem is paired with its optimal reasoning chain—defined as the shortest chain that leads to a correct answer. This dataset will directly teach the final model the skill of adaptive thinking.

**Data Curation Pipeline.** For each problem $x_i$ in our training set, we execute the following pipeline:

1. Reasoning Generation: We input $x_i$ into every model $M_\alpha$ in our sampled spectrum. Each model generates a reasoning chain $\text{CoT}_i^\alpha$ and a predicted answer $y_i^\alpha$.

2. Answer Verification: We verify the correctness of each predicted answer $y_i^\alpha$ against the ground-truth answer $y_i$ using a task-specific criterion (e.g., exact match for mathematical answers, predefined evaluation metrics for other reasoning tasks).

3. Optimal Chain Selection: From all models that produced the correct answer ($y_i^\alpha = y_i$), we select the reasoning chain $\mathrm{CoT}_i^{\alpha^*}$ from the model with the largest value of $\alpha$ (i.e., the model biased towards the shortest chains). Formally: $\alpha^* = \max\{\alpha \in S \mid y_i^\alpha = y_i\}$, where $S$ is the set of sampled $\alpha$ values. The corresponding $\mathrm{CoT}_i^{\alpha^*}$ is deemed the optimal adaptive chain for problem $x_i$.

4. Data Assignment: The tuple $(x_i, y_i, \mathrm{CoT}_i^{\alpha^*})$ is added to the new adaptive training dataset $\mathcal{D}_{\mathrm{adaptive}}$.

**Handling Edge Cases.** If no model in the spectrum produces the correct answer, the problem $x_i$ is excluded from $\mathcal{D}_{\mathrm{adaptive}}$. This ensures the quality of the training data. If multiple models with the same $\alpha$ value produce the correct answer, the shortest generated chain among them is selected, providing a further refinement.

**Theoretical Justification and Outcome.** This selection protocol is a data-driven implementation of the motivation described in Section 3.1. For each problem, it identifies the operational point near the "elbow" of the sigmoid curve—the point where accuracy is achieved with minimal computational cost. The resulting dataset $\mathcal{D}_{\mathrm{adaptive}}$ no longer contains chains of a fixed length. Instead, it is a collection of difficulty-labeled examples by proxy; the length of the chain $\mathrm{CoT}_i^{\alpha^*}$ represents the complexity of the problem $x_i$.

By learning from these optimal (problem, chain) pairs, a model can be trained to emulate this optimal selection process dynamically at inference time. The final dataset $\mathcal{D}_{\mathrm{adaptive}} = \{(x_i, y_i, \mathrm{CoT}_i^{\mathrm{opt}})\}_{i=1}^M$ is the key resource for training our adaptive model in the final step.

## 3.5 STEP 4: TRAINING THE ADAPTIVE MODEL

The final and crucial step of the DART framework is to distill the collective knowledge of the model spectrum and the optimal adaptive policy embodied in $\mathcal{D}_{\mathrm{adaptive}}$ into a single, efficient, and standalone model, denoted as $M_{\mathrm{adaptive}}$. This model is designed to intrinsically learn the mapping from problem difficulty to reasoning length, enabling it to generate the minimally sufficient chain-of-thought autonomously during inference, without relying on the cumbersome process of generating multiple chains from a spectrum of models.

**Training Objective.** We initialize $M_{\mathrm{adaptive}}$ from the same pre-trained checkpoint as the base and distilled models. The model is then fine-tuned on the curated dataset $\mathcal{D}_{\mathrm{adaptive}}$ using standard supervised fine-tuning (SFT). The learning objective is to maximize the likelihood of generating the optimal reasoning chain $\mathrm{CoT}_i^{\mathrm{opt}}$ and the correct answer $y_i$ given the input question $x_i$:

$$\mathcal{L}_{\mathrm{adaptive}} = - \sum_{(x,y,\mathrm{CoT}^{\mathrm{opt}}) \in \mathcal{D}_{\mathrm{adaptive}}} \sum_t \log P(w_t | x, w_{<t}) \tag{4}$$

where $w_t$ is the $t$-th token in the sequence $[\mathrm{CoT}^{\mathrm{opt}}, y]$.

**The Essence of Adaptive Learning.** The key innovation is what the model learns from $\mathcal{D}_{\mathrm{adaptive}}$ : For simple problems that required only a short $\mathrm{CoT}^{\mathrm{opt}}$ in the dataset, the model learns to generate concise reasoning. For complex problems that required a longer $\mathrm{CoT}^{\mathrm{opt}}$, it learns to deploy more elaborate reasoning steps.

Unlike the model fusion step which controls behavior externally via the $\alpha$ parameter, $M_{\mathrm{adaptive}}$ learns to internalize the decision-making process. It does not merely imitate a fixed style but learns a spectrum of behaviors and, crucially, the conditional logic of when to apply each style. The training process teaches the model to approximate the function $f(x) = \mathrm{CoT}^{\mathrm{opt}}$, effectively compressing the entire model spectrum and selection pipeline into a single network.

## 4 EXPERIMENTS

### 4.1 EXPERIMENTAL SETUP

**Backbone Reasoning Models.** We conduct experiments on four representative open-source large language models with strong reasoning capabilities: the Qwen3 (Yang et al., 2025) series (4B-Thinking-2507 [1], 8B, and 14B parameter versions) and DeepSeek-R1-Distill-Qwen-7B (Guo et al., 2025). These models were selected for their native chain-of-thought reasoning capabilities and represent diverse architectural approaches to reasoning in language models.

**Datasets.** We evaluate our method on five challenging mathematical reasoning benchmarks: GSM8K (Cobbe et al., 2021), MATH-500 (Lightman et al., 2023), AMC23 (Committees, 2023), OLYMPAID (He et al., 2024), and AIME25 (Committees, 2025). These datasets span a wide range of difficulty levels, providing comprehensive evaluation of reasoning capabilities.

**Metrics.** We employ three key metrics to comprehensively evaluate performance: (1) **Pass@1**: Problem-solving accuracy (primary quality metric); (2) **ACT** (Average chain-of-thought Tokens): Average number of tokens in the reasoning chain (efficiency metric); (3) **AAT** (Average Answer Tokens): Average total tokens in model output including reasoning and final answer (overall efficiency metric).

**Baselines.** We also present a comprehensive comparison between our proposed method and several state-of-the-art reasoning optimization approaches. We categorize the baseline approaches into the following groups: (1) **Prompt-based methods**: These rely on prompt engineering without parameter updates. We select CoD (Xu et al., 2025) and TALE-EP (Han et al., 2024) as our baselines. (2) **RL-based methods**: These leverage reinforcement learning, typically with reward signals derived from task-specific objectives or human feedback, to optimize reasoning performance. We select Z1 (Yu et al., 2025), DAST (Shen et al., 2025), AdaptThink (Zhang et al., 2025a) as our baselines. (3) **SFT-based methods**: These utilize supervised fine-tuning on curated datasets to learn reasoning patterns. We select AutoL2S (Luo et al., 2025) as our baseline.

**Implementation Details.** We obtained Long CoT data from DeepSeek-R1-Distill(tuanha1305, 2025) dataset, which contains problem-solution pairs with detailed reasoning chains, and the short CoT compressed from the long chains using DeepSeek-R1 (Guo et al., 2025) model with specific compression prompts. The complete prompt is provided in Appendix B.1. We use Qwen3-14B as our base model to train a distilled version for model fusion. The training details can be refered in Appendix B.2 and the model fusion details can be refered in Appendix B.3. Adaptive training data is constructed from the training splits of GSM8K (Cobbe et al., 2021) and MATH (Hendrycks et al., 2021) datasets using our proposed curation pipeline. Please refer to Appendix B.4 for adaptive training details.

### 4.2 MAIN RESULTS

Table 1 presents the comprehensive comparison across all datasets and model scales. Our proposed DART method demonstrates superior efficiency-accuracy trade-offs compared to all baselines.

**Consistent Efficiency Gains.** DART consistently reduces computational cost across all model scales and datasets. The efficiency gains vary with problem difficulty: on simpler datasets like GSM8K, token usage is reduced by 34.0%–81.2%; while on highly challenging benchmarks AIME25, the method still achieves substantial savings up to 34.2%, demonstrating its ability to adaptively allocate more tokens to harder problems. This validates our core hypothesis that optimal reasoning length should adapt to problem difficulty.

**Accuracy Preservation.** DART maintains competitive accuracy across most settings and, notably, improves Pass@1 accuracy in several cases—such as on MATH-500, AMC23 and AIME25 (all Qwen3 scales)—while

---

[1]Throughout this paper, all Qwen3-4B results refer to the Qwen3-4B-Thinking-2507 variant.

Table 1: Performance comparison of different methods across datasets.

| Method | GSM8K | | | MATH-500 | | | AMC23 | | | OLYMPAID | | | AIME25 | | |
|---|---|---|---|---|---|---|---|---|---|---|---|---|---|---|---|
| | Pass@1 | ACT | AAT | Pass@1 | ACT | AAT | Pass@1 | ACT | AAT | Pass@1 | ACT | AAT | Pass@1 | ACT | AAT |
| **Qwen3-4B** | | | | | | | | | | | | | | | |
| Vanilla | **95.2** | 1253.31 | 1557.70 | 96.0 | 5894.34 | 6699.73 | 97.5 | 10524.80 | 11362.50 | 72.9 | 12298.35 | 14863.27 | 70.0 | 16379.95 | 21496.90 |
| *Prompt-based* | | | | | | | | | | | | | | | |
| CoD | 94.9 (-0.3) | 855.34 (-31.8%) | 955.29 (-38.7%) | 95.2 (-0.8) | 3676.47 (-37.6%) | 4254.38 (-36.5%) | 97.5 (+0.0) | 8535.40 (-18.9%) | 9256.63 (-18.5%) | 72.3 (-0.6) | 10065.13 (-18.1%) | 12004.26 (-19.2%) | 73.3 (+3.3) | 16226.83 (-0.9%) | 20284.77 (-5.6%) |
| TALE-EP | 94.6 (-0.6) | 4294.88 (+242.7%) | 5100.53 (+227.4%) | 94.8 (-1.2) | 6533.66 (+10.8%) | 8040.52 (+20.0%) | 97.5 (+0.0) | 10773.84 (+2.4%) | 12348.10 (+8.7%) | 74.2 (+1.3) | 11947.04 (-2.9%) | 14661.11 (-1.4%) | 76.7 (+6.7) | 17718.96 (+8.2%) | 21450.47 (-0.2%) |
| *SFT-based* | | | | | | | | | | | | | | | |
| **DART(Ours)** | 93.9 (-1.3) | **401.13 (-68.0%)** | **596.37 (-61.7%)** | 96.4 (+0.4) | 3391.92 (-42.5%) | 3981.97 (-40.6%) | 100.0 (+2.5) | 6661.93 (-36.7%) | 7379.20 (-35.1%) | 72.0 (-0.9) | 8758.18 (-28.8%) | 10271.33 (-30.9%) | 80.0 (+10.0) | 16071.10 (-1.9%) | 17513.30 (-18.5%) |
| **Qwen3-8B** | | | | | | | | | | | | | | | |
| Vanilla | **95.7** | 1887.56 | 2214.61 | 94.4 | 4543.18 | 5309.38 | 92.5 | 8001.18 | 9436.85 | 68.6 | 9850.64 | 11257.47 | 56.7 | 15110.00 | 19063.27 |
| *Prompt-based* | | | | | | | | | | | | | | | |
| CoD | 95.6 (-0.1) | **498.38 (-73.6%)** | **598.36 (-73.0%)** | 94.6 (+0.2) | 2881.84 (-36.6%) | 3539.03 (-33.3%) | 95.0 (+2.5) | 5949.68 (-25.6%) | 6697.40 (-29.0%) | 67.0 (-1.6) | 7713.47 (-21.7%) | 9008.56 (-20.0%) | 63.3 (+6.6) | 14399.34 (-4.7%) | 15984.17 (-16.2%) |
| TALE-EP | 93.5 (-2.2) | 1148.60 (-39.1%) | 1421.65 (-35.8%) | 94.6 (+0.2) | 3617.01 (-20.4%) | 4580.30 (-13.7%) | 97.5 (+5.0) | 5987.89 (-25.2%) | 7438.82 (-21.2%) | 62.4 (-6.2) | 8637.41 (-12.3%) | 10268.04 (-8.8%) | 60.0 (+3.3) | 15473.76 (+2.4%) | 19111.93 (+0.3%) |
| *SFT-based* | | | | | | | | | | | | | | | |
| **DART(Ours)** | 95.1 (-0.6) | 983.87 (-47.9%) | 1262.60 (-43.0%) | 95.6 (+1.2) | 3321.36 (-26.9%) | 3985.53 (-24.9%) | 97.5 (+5.0) | 5204.95 (-34.9%) | 5996.90 (-36.5%) | 68.0 (-0.6) | 7468.58 (-24.2%) | 8549.27 (-24.1%) | 66.7 (+10.0) | 12610.43 (-16.5%) | 13560.50 (-28.9%) |
| **Qwen3-14B** | | | | | | | | | | | | | | | |
| Vanilla | 95.8 | 1399.16 | 1709.04 | 94.8 | 4075.58 | 4776.44 | 97.5 | 6691.5 | 7544.35 | 70.5 | 8695.07 | 10086.94 | 63.3 | 13324.23 | 16878.13 |
| *Prompt-based* | | | | | | | | | | | | | | | |
| CoD | 95.9 (+0.1) | 535.22 (-61.7%) | 617.18 (-63.9%) | 95.4 (+0.6) | 2532.20 (-37.9%) | 2988.71 (-37.4%) | 97.5 (+0.0) | 4888.58 (-26.9%) | 5563.3 (-26.3%) | 70.2 (-0.3) | 6837.84 (-21.4%) | 7685.47 (-23.8%) | 66.7 (+3.4) | 12811.03 (-3.9%) | 15066.03 (-10.7%) |
| TALE-EP | 92.8 (-3.0) | **94.10 (-93.3%)** | **169.78 (-90.1%)** | 79.2 (-15.6) | 368.34 (-91.0%) | 655.64 (-86.3%) | 70.0 (-27.5) | 1015.75 (-84.8%) | 1424.80 (-81.1%) | 48.0 (-22.5) | **764.35 (-91.2%)** | 2072.40 (-79.5%) | 16.7 (-46.6) | 950.93 (-92.9%) | 1617.63 (-90.4%) |
| *SFT-based* | | | | | | | | | | | | | | | |
| **DART(Ours)** | **96.4 (+0.6)** | 923.04 (-34.0%) | 1165.95 (-31.8%) | 96.4 (+1.6) | 3161.88 (-22.4%) | 3748.81 (-21.5%) | 100.0 (+2.5) | 4831.25 (-27.8%) | 5601.65 (-25.8%) | 70.4 (-0.1) | 7165.85 (-17.6%) | 8206.90 (-18.6%) | 70.0 (+6.7) | 11779.24 (-11.6%) | 13446.47 (-20.3%) |
| **DeepSeek-R1-Distill-Qwen-7B** | | | | | | | | | | | | | | | |
| Vanilla | 90.2 | 895.19 | 1007.26 | 91.0 | 2847.29 | 3385.94 | 90.0 | 5288.93 | 5789.63 | 57.8 | 6933.88 | 8003.48 | 36.7 | 13276.79 | 15060.97 |
| *Prompt-based* | | | | | | | | | | | | | | | |
| CoD | 83.6 (-6.6) | 184.62 (-79.4%) | **312.68 (-69.0%)** | 86.2 (-4.8) | **1587.01 (-44.3%)** | 1902.69 (-43.8%) | 87.5 (-2.5) | 4383.23 (-17.1%) | 4723.45 (-18.4%) | 55.6 (-2.2) | **4792.10 (-30.9%)** | 5407.04 (-32.4%) | 40.0 (+3.3) | 11009.93 (-17.1%) | 11495.60 (-23.7%) |
| TALE-EP | 90.1 (-0.1) | 927.17 (+3.6%) | 985.47 (-2.2%) | 62.6 (-28.4) | 2809.62 (-1.3%) | 2860.74 (-15.5%) | 70.0 (-20.0) | 6459.59 (+22.1%) | 6208.88 (+7.2%) | 45.5 (-10.1) | 7583.87 (+58.3%) | 8113.98 (+50.1%) | 23.3 (-13.4) | 12607.29 (-5.0%) | 12261.93 (-18.6%) |
| *RL-Based* | | | | | | | | | | | | | | | |
| Z1 | 89.3 (-0.9) | - | 591.38 (-41.3%) | 74.6 (-16.4) | - | 1437.84 (-57.5%) | 37.5 (-52.5) | - | 2657.4 (-54.1%) | 37.8 (-20.0) | - | 2317.77 (-71.0%) | 10.0 (-26.7) | - | **3986.63 (-73.5%)** |
| DAST | 91.6 (+1.4) | 860.06 (-3.9%) | 976.15 (-3.1%) | 92.6 (+1.6) | 3123.93 (+9.7%) | 3666.38 (+8.3%) | 90.0 (+0.0) | 4657.95 (-11.9%) | 5820.28 (+0.5%) | 60.9 (+3.1) | 6860.06 (-1.1%) | 7820.03 (-2.3%) | 33.3 (-3.4) | 12727.72 (-4.1%) | 13931.33 (-7.5%) |
| AdaptThink | 84.7 (-5.5) | 174.97 (-80.5%) | 457.38 (-54.6%) | 86.0 (-5.0) | 2226.04 (-21.8%) | 2716.27 (-19.8%) | 82.5 (-7.5) | 4882.35 (-7.7%) | 5370.65 (-7.2%) | 53.9 (-3.9) | 6538.94 (-5.7%) | 7480.51 (-6.5%) | 36.7 (+0.0) | 13891.56 (+4.6%) | 16235.87 (+7.8%) |
| *SFT-Based* | | | | | | | | | | | | | | | |
| AutoL2S | 91.0 (+0.8) | - | 481.34 (-52.2%) | 77.6 (-13.4) | - | **1326.59 (-60.8%)** | 47.5 (-52.5) | - | 3639.50 (-37.1%) | 41.9 (-15.9) | - | 3011.38 (-62.4%) | 10.0 (-26.7) | - | 5303.93 (-64.8%) |
| **DART(Ours)** | 89.1 (-1.1) | **168.00 (-81.2%)** | 358.40 (-64.4%) | 88.6 (-2.4) | 1853.43 (-34.9%) | 2355.73 (-30.4%) | 90.0 (+0.0) | 3460.26 (-34.6%) | 4518.10 (-22.0%) | 55.4 (-2.4) | 5076.86 (-26.8%) | 6406.84 (-19.9%) | 36.7 (+0.0) | 8729.72 (-34.2%) | 9974.80 (-33.8%) |

simultaneously reducing computational cost. This result suggests that adaptive termination can eliminate redundant or counterproductive reasoning steps.

## 4.3 COMPARISON TO PRIOR WORK

As shown in Table 1, our experimental analysis reveals a significant methodological divide in existing adaptive reasoning approaches. RL-based methods (Z1, DAST, AdaptThink) demonstrate strong model specialization but are predominantly developed and evaluated exclusively on DeepSeek-R1-Distill-Qwen-7B, with no available implementations for Qwen3 series models. Conversely, prompt-based approaches (CoD, TALE-EP) offer broader model compatibility but face fundamental limitations in accuracy-efficiency trade-offs. This landscape highlights the unique positioning of our method in overcoming these constraints.

**Superior Model Compatibility and Flexibility.** Unlike RL-based methods that struggle with Qwen3's extensive RLHF training, DART demonstrates remarkable architectural agnosticism. It achieves consistent improvements across all tested models (Qwen3-4B/8B/14B and DeepSeek-R1), proving its adaptability to diverse model architectures. While prompt-based methods like CoD and TALE-EP show wider model adaptability than RL-based techniques, they incur substantial accuracy costs across all model sizes. For

instance, CoD exhibits accuracy degradation up to 6.6 on DeepSeek-R1-Distill-Qwen-7B, while TALE-EP shows similar patterns on all tested models.

**Enhanced Generalization Capabilities.** DART exhibits superior cross-dataset generalization compared to both RL-based and prompt-based approaches. Despite being trained only on GSM8K and MATH, it maintains strong performance on out-of-distribution benchmarks (AMC23, OLYMPAID, AIME25). Prompt-based methods like CoD and TALE-EP show inconsistent generalization patterns, with significant accuracy variations across datasets. RL-based methods demonstrate even narrower generalization, often failing to transfer adaptive policies beyond their training distribution. DART's supervised learning framework appears to capture more fundamental reasoning-length principles that translate better across problem types.

**Training Stability and Practical Deployment.** The supervised learning paradigm of DART offers significant advantages over RL-based methods in terms of training stability and reproducibility. While RL methods require careful reward engineering and suffer from optimization instability, DART employs standard fine-tuning procedures that yield consistent results. In contrast to methods needing intricate prompt design, DART's trained approach ensures reliable inference performance without manual intervention. This makes it ideal for production environments that demand consistency.

The comprehensive comparison shows that DART achieves what previous methods cannot: effective adaptive reasoning across diverse model architectures with robust generalization and practical deployability.

### 4.4 GENERALIZATION TO NON-MATHEMATICAL REASONING TASKS

To further validate the generalizability of the DART framework, we evaluate its performance on three challenging non-mathematical reasoning benchmarks: GPQA (Rein et al., 2024), a demanding dataset of PHD-level questions in physics, biology, and chemistry; LogiQA (Liu et al., 2020), a dataset for logical reasoning; and CommonsenseQA (Talmor et al., 2019), which tests commonsense reasoning. Crucially, the DART models evaluated here are the same models trained only on the mathematical reasoning datasets (GSM8K and MATH)–without any domain-specific fine-tuning–assessing their zero-shot cross-domain transfer capability.

As shown in Table 2, DART demonstrates robust cross-domain generalization with consistent efficiency improvements and accuracy preservation across all domains. On GPQA, DART reduces ACT by 16.6%-46.9% while improving Pass@1 accuracy by up to 5.1 (Qwen3-14B); on LogiQA, ACT reductions of 20.9%-48.5% are achieved while maintaining comparable accuracy; and on CommonsenseQA, efficiency gains of 7.0%-49.2% are accompanied by accuracy improvements of up to 10.6 (DeepSeek-R1-Distill-Qwen-7B). This consistent pattern across scientific, logical, and commonsense reasoning tasks demonstrates that DART learns fundamental principles of reasoning sufficiency rather than domain-specific patterns, effectively establishing its value as a general-purpose adaptive reasoning framework.

Table 2: Cross-domain generalization performance of DART on non-mathematical reasoning benchmarks. All DART models were trained only on mathematical reasoning datasets (GSM8K and MATH).

| Method | GPQA | | | LogiQA | | | CommonsenseQA | | |
|---|---|---|---|---|---|---|---|---|---|
| | Pass@1 | ACT | AAT | Pass@1 | ACT | AAT | Pass@1 | ACT | AAT |
| **Qwen3-4B** | | | | | | | | | |
| Vanilla | $63.1_{+0.0}$ | $8048.96_{-0.0\%}$ | $9070.81_{-0.0\%}$ | $81.6_{+0.0}$ | $3572.32_{-0.0\%}$ | $4268.86_{-0.0\%}$ | $71.6_{+0.0}$ | $2037.42_{-0.0\%}$ | $2536.90_{-0.0\%}$ |
| DART(Ours) | $66.2_{+3.1}$ | $6710.46_{-16.6\%}$ | $7558.13_{-16.7\%}$ | $81.4_{-0.2}$ | $2825.63_{-20.9\%}$ | $3368.78_{-21.1\%}$ | $77.0_{+5.4}$ | $1895.72_{-7.0\%}$ | $2223.47_{-12.4\%}$ |
| **Qwen3-8B** | | | | | | | | | |
| Vanilla | $58.1_{+0.0}$ | $8632.48_{-0.0\%}$ | $9660.64_{-0.0\%}$ | $80.2_{+0.0}$ | $2533.89_{-0.0\%}$ | $3127.05_{-0.0\%}$ | $75.6_{+0.0}$ | $1141.44_{-0.0\%}$ | $1621.51_{-0.0\%}$ |
| DART(Ours) | $57.8_{-0.3}$ | $5980.81_{-30.7\%}$ | $6790.24_{-29.7\%}$ | $80.7_{+0.5}$ | $1689.21_{-33.3\%}$ | $2222.22_{-28.9\%}$ | $77.4_{+1.8}$ | $807.28_{-29.3\%}$ | $1193.31_{-26.4\%}$ |
| **Qwen3-14B** | | | | | | | | | |
| Vanilla | $60.6_{+0.0}$ | $7161.52_{-0.0\%}$ | $8083.38_{-0.0\%}$ | $83.7_{+0.0}$ | $1970.96_{-0.0\%}$ | $2544.78_{-0.0\%}$ | $75.4_{+0.0}$ | $884.82_{-0.0\%}$ | $1289.27_{-0.0\%}$ |
| DART(Ours) | $65.7_{+5.1}$ | $5286.08_{-26.2\%}$ | $6123.59_{-24.2\%}$ | $84.5_{+0.8}$ | $1476.42_{-25.1\%}$ | $2012.44_{-20.9\%}$ | $77.4_{+2.0}$ | $663.28_{-25.0\%}$ | $1028.92_{-20.2\%}$ |
| **DeepSeek-R1-Distill-Qwen-7B** | | | | | | | | | |
| Vanilla | $40.9_{+0.0}$ | $6166.27_{-0.0\%}$ | $7158.12_{-0.0\%}$ | $48.7_{+0.0}$ | $1633.06_{-0.0\%}$ | $2196.57_{-0.0\%}$ | $40.0_{+0.0}$ | $885.90_{-0.0\%}$ | $1322.53_{-0.0\%}$ |
| DART(Ours) | $43.9_{+3.0}$ | $3275.42_{-46.9\%}$ | $5953.13_{-16.8\%}$ | $48.7_{+0.0}$ | $840.55_{-48.5\%}$ | $1474.76_{-32.9\%}$ | $50.6_{+10.6}$ | $449.62_{-49.2\%}$ | $953.00_{-27.9\%}$ |

## 4.5 FURTHER ANALYSIS

### 4.5.1 VALIDATION OF MODEL FUSION EFFECTIVENESS

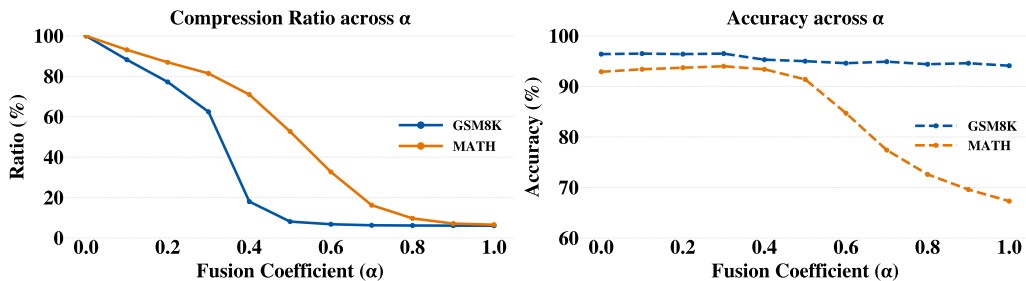

Figure 3: Effect of fusion coefficient ($\alpha$) on compression ratio and accuracy.

In practice, we sample 11 values of $\alpha$ (e.g., $\alpha \in \{0, 0.1, ..., 0.9, 1\}$) to create a discrete but dense ensemble of models $\{M_{\alpha_1}, M_{\alpha_2}, ..., M_{\alpha_{11}}\}$. This ensemble allows us to approximate the continuum and explore the full range of reasoning lengths.

As shown in Fig. 3, we observe that the average length of the generated CoT is a smooth, monotonically decreasing function of $\alpha$. This confirms that our fusion method successfully creates a controllable knob for adjusting reasoning complexity without additional training. Furthermore, the accuracy on reasoning benchmarks reveals a more nuanced relationship: as $\alpha$ increases from 0 to 1, accuracy initially shows a slight improvement before entering a declining phase. This non-monotonic behavior demonstrates that longer reasoning chains do not invariably lead to higher accuracy; beyond a certain point, excessive verbosity can be counterproductive. The fusion spectrum thus effectively samples different points on the efficiency-accuracy Pareto frontier, providing an efficient and elegant solution for generating the multi-length reasoning data required for the next step of our pipeline.

### 4.5.2 IMPACT OF DATA CURATION REPETITION

The adaptive training data generation process involves multiple passes through our model spectrum to identify optimal reasoning chains. Fig. 4 presents the effect of repetition frequency on final model performance:

**Consistency Across Repetitions.** Performance remains stable across passes. For Qwen3-4B on MATH-500, accuracy stays high (95.4%–96.4%) with only minor ACT variation (3355.4–3696.0). Similarly, for Qwen3-8B on GSM8K maintains 95% accuracy across passes. This indicates that a single model pass (Pass@1) is sufficient to collect high-quality adaptive examples.

**Efficiency-Accuracy Trade-offs.** Additional passes bring limited and inconsistent efficiency gains. For example, Qwen3-4B on GSM8K shows ACT reduction from 512.93 to 401.13 over three passes, but this trend does not hold on harder datasets. This suggests diminishing returns beyond the first pass, making Pass@1 the most practical choice.

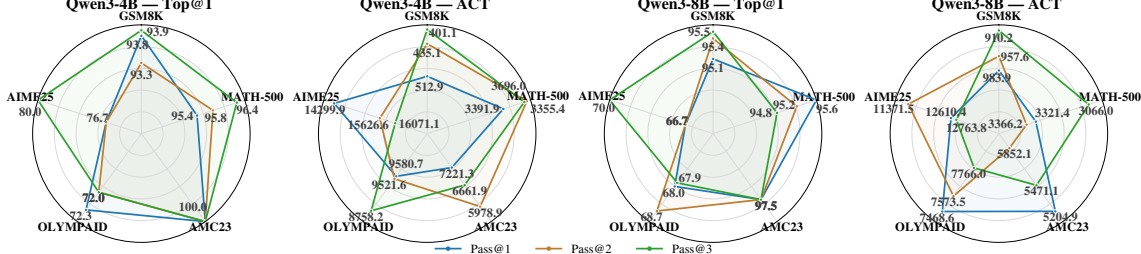

Figure 4: Effect of data curation iterations on model performance across different datasets.

### 4.5.3 SENSITIVITY TO $\alpha$ SAMPLING DENSITY.

We compare loose (5 points), middle (10 points), and dense (20 points) $\alpha$ sampling strategies in Table 3:

**Density-Accuracy Relationship.** Contrary to expectations, denser sampling does not consistently improve accuracy. On Qwen3-8B/MATH-500, loose sampling achieves the highest accuracy (96.6%), while dense sampling gives comparable results (96.0%). A sparse sampling (5–10 points) thus sufficiently captures key reasoning behaviors.

**Efficiency Optimization.** Efficiency, however, improves with density: on Qwen3-4B/GSM8K, dense sampling yields the lowest ACT (421.67) versus loose (986.11) and middle (725.80), though with a slight accuracy drop (93.6% vs. 95.5%). Loose sampling favors accuracy; dense sampling favors token reduction.

Based on these findings, we recommend middle-density sampling (10 $\alpha$ points) as the default configuration, balancing computational cost during data curation with final model performance. This density provides sufficient granularity to identify near-optimal reasoning lengths without excessive computational overhead.

Table 3: Impact of $\alpha$ sampling density on model performance across different datasets.

| Sampling Density | GSM8K | | | MATH-500 | | | AMC23 | | | OLYMPAID | | | AIME25 | | |
|---|---|---|---|---|---|---|---|---|---|---|---|---|---|---|---|
| | Pass@1 | ACT | AAT | Pass@1 | ACT | AAT | Pass@1 | ACT | AAT | Pass@1 | ACT | AAT | Pass@1 | ACT | AAT |
| **Qwen3-4B** | | | | | | | | | | | | | | | |
| Loose | 95.5 | 986.11 | 1230.04 | 95.6 | 3613.50 | 4248.40 | 100.0 | 6224.53 | 6929.80 | 71.4 | 9322.56 | 11327.69 | 76.7 | 14323.03 | 16386.83 |
| Middle | 94.7 | 725.80 | 952.60 | 95.6 | 3344.17 | 3893.69 | 100.0 | 6824.88 | 7590.55 | 70.8 | 9151.90 | 10623.66 | 70.0 | 14345.74 | 17057.70 |
| Dense | 93.6 | 421.67 | 602.24 | 96.2 | 3637.91 | 4166.00 | 100.0 | 5803.83 | 6409.65 | 71.1 | 9392.36 | 10385.10 | 76.7 | 15556.56 | 18016.97 |
| **Qwen3-8B** | | | | | | | | | | | | | | | |
| Loose | 95.7 | 1114.57 | 1397.32 | 96.6 | 3377.11 | 3974.48 | 97.5 | 5503.95 | 6282.48 | 67.6 | 7881.85 | 8952.07 | 70.0 | 12986.69 | 14652.53 |
| Middle | 95.7 | 1095.47 | 1408.25 | 95.4 | 3271.15 | 3911.19 | 100.0 | 6412.08 | 7243.20 | 67.9 | 7744.78 | 8797.41 | 63.3 | 12235.40 | 15225.40 |
| Dense | 94.8 | 682.98 | 924.61 | 96.0 | 3572.20 | 4172.57 | 95.0 | 5719.13 | 6323.32 | 68.6 | 7722.53 | 8741.07 | 70.0 | 12546.3 | 13584.63 |

### 4.5.4 EXPLICIT VALIDATION OF DIFFICULTY-AWARE ADAPTATION.

To explicitly validate DART's ability to adapt reasoning length based on problem difficulty, we conduct a fine-grained analysis using the MATH-500 dataset with explicit difficulty ratings (level 1-5). As illustrated in Figure 5, evaluating DART (Qwen3-4B) across these levels reveals a clear difficulty-aware pattern: for easier problems (levels 1-2), high compression ratios (56.9% and 56.4%) are achieved while maintaining near-perfect accuracy (100.0% and 96.7%). As difficulty increases to level 3, compression slightly decreases to 55.2% to preserve accuracy (98.1%), and for the hardest problems (levels 4-5), compression further drops to 47.1% and 34.0%, prioritizing accuracy over efficiency.

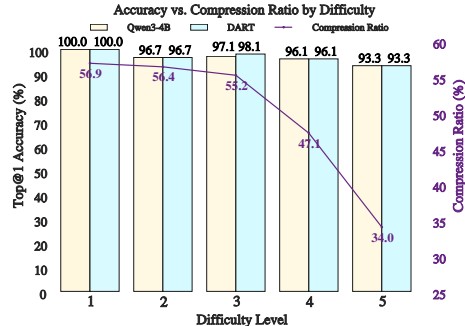

Figure 5: Accuracy vs. compression ratio across different difficulty levels on MATH-500 dataset (Qwen3-4B).

This graded adaptation demonstrates that DART successfully internalizes the mapping from problem difficulty to optimal reasoning length. Rather than applying uniform compression, it dynamically allocates more computational resources to harder problems while truncating reasoning on simpler ones, confirming its ability to make nuanced decisions based on problem complexity.

## 5 CONCLUSION

In this work, we presented DART, a supervised framework for difficulty-adaptive reasoning truncation that allows LLMs to dynamically adjust their thinking length according to problem complexity. By distilling concise reasoning patterns, interpolating them into a continuum of reasoning styles, and curating optimal training signals, it learns when to stop thinking. DART realizes significant reasoning truncation and speedup without sacrificing accuracy, paving the way toward more efficient and sustainable LLMs.

## 6 ETHICS STATEMENT

All datasets used in this study are publicly available mathematical reasoning benchmarks (GSM8K, MATH-500, AMC23, OLYMPAID, AIME25, DeepSeek-R1-Distill) that contain no personally identifiable information or sensitive content. The data generated during research consists solely of mathematical problems and corresponding reasoning processes, posing no ethical risks.

The large language models used in experiments (including Qwen3 series and DeepSeek-R1) are open-source models used in compliance with their respective licenses. This research focuses on improving reasoning efficiency and does not involve generating harmful or misleading content.

The proposed DAST methodology aims to reduce computational resource consumption, aligning with sustainable development principles. All experiments were conducted within an academic research framework with no potential harm to any individuals or groups.

## 7 REPRODUCIBILITY STATEMENT

We are committed to the principle of reproducibility. While the source code and models are not publicly released at this time, they may be made available upon reasonable request to the corresponding author for academic, non-commercial purposes. To facilitate replication, key implementation details and all critical hyperparameters are detailed in Appendix.

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

## A    STATEMENT ON THE USE OF LARGE LANGUAGE MODELS (LLMS)

The authors utilized a Large Language Model (DeepSeek API) exclusively to assist in the writing process of this manuscript. The use of the LLM was strictly limited to:

1. Language polishing and grammar checking for improved fluency and academic tone.

2. Rewriting and restructuring of initial drafts composed by the authors to enhance logical flow and readability.

3. Suggesting terminology to ensure precise expression in specific contexts.

The LLM served solely as a supplementary tool. All generated content was rigorously reviewed, critically evaluated, and substantially modified by the authors, who assume full responsibility for the entire work's factual accuracy, data integrity, academic arguments, and conclusions.

## B    IMPLEMENTATION DETAILS

### B.1    PROMPT TEMPLATE OF SHORT CoT DATA DISTILLATION

The full prompt for prompt template of short CoT data distillation is shown in Figure 6.

---

**Prompt templates of short CoT data distillation.**

Please strictly follow the following requirements to condense the text:
1. Preserve the original reasoning logic and key steps.
2. Remove redundant descriptions, repetitions, and irrelevant details.
3. Ensure the simplified text can independently complete the same reasoning task.
4. Compress the text to its most concise form. Return only the simplified text without additional explanations.

The input consists of: "input" (a math problem), "reasoning_content" (the reasoning process based on the problem) and "content" (the problem's answer output).

Please simplify the "reasoning_content" part.

Input text: {*text*}

Condensed text (output directly):

---

Figure 6: Prompt template of short CoT data distillation.

## B.2    DETAILS OF DISTILLED MODEL TRAINING

We fine-tune the `Qwen3-14B` model using the LLaMA-Factory (Zheng et al., 2024) framework, employing a full-parameter fine-tuning approach. The training process is optimized with the DeepSpeed ZeRO Stage 3 strategy. The training data consists of 30,000 math samples from the DeepSeek-R1-Distill(tuanha1305, 2025) dataset. We set a cutoff length of 32,768 tokens for the sequences.

The model is fine-tuned for 3 epochs with the following hyperparameters:

- **Cutoff length:** 32,768
- **Max samples:** 30,000
- **Batch size:** 1 (with a gradient accumulation of 2)
- **Learning rate:** $5 \times 10^{-6}$ with a cosine schedule and a warmup ratio of 0.1
- **Precision:** `bf16`
- **Validation split:** 10% of the training data
- **Evaluation strategy:** every 200 steps

All experiments are conducted with the `overwrite_cache=true` option and utilize 16 parallel workers for data preprocessing. The resulting models are directly used for downstream evaluation without any additional tuning.

## B.3    DETAILS OF MODEL FUSION

We use Qwen3-14B as our base model to train a distilled version for model fusion. The fusion coefficient ($\alpha$) we used can be referred in Table B.3.

## B.4    DETAILS OF ADAPTIVE MODEL TRAINING

We perform adaptive fine-tuning on the model using the LLaMA-Factory (Zheng et al., 2024) framework with Low-Rank Adaptation (LoRA) (Hu et al., 2022). The LoRA configuration targets all linear layers (`lora_target: all`) with a rank of 256 and an alpha of 16. The training source question data comprises

Table 4: Fusion coefficient ($\alpha$) used for model fusion.

| Dataset | Alpha | Accuracy | Avg Tokens | | Ratio |
|---|---|---|---|---|---|
| | | | ACT | AAT | |
| | 0 (Qwen3-14B) | 96.6 | 1325.15 | 1632.77 | 100.00% |
| | 0.05 | 96.5 | 1288.49 | 1589.59 | 97.23% |
| | 0.1 | 96.4 | 1206.89 | 1496.72 | 91.08% |
| | 0.125 | 96.6 | 1141.44 | 1425.52 | 86.14% |
| | 0.15 | 96.5 | 1108.54 | 1400.72 | 83.65% |
| | 0.175 | 96.4 | 1002.43 | 1263.96 | 75.65% |
| | 0.2 | 96.4 | 988.16 | 1252.73 | 74.57% |
| | 0.225 | 96.7 | 957.73 | 1199.52 | 72.27% |
| | 0.24 | 96.6 | 951.57 | 1193.46 | 71.81% |
| | 0.25 | 96.5 | 846.61 | 1042.58 | 63.89% |
| | 0.275 | 96.5 | 741.08 | 918.67 | 55.92% |
| | 0.3 | 96.6 | 703.36 | 882.84 | 53.08% |
| | 0.325 | 96.4 | 668.81 | 844.13 | 50.47% |
| | 0.35 | 96.5 | 647.15 | 825.05 | 48.84% |
| GSM8K | 0.375 | 96.5 | 575.50 | 743.11 | 43.43% |
| | 0.4 | 96.6 | 487.18 | 657.17 | 36.76% |
| | 0.42 | 96.2 | 409.42 | 578.93 | 30.90% |
| | 0.44 | 96.2 | 360.85 | 530.66 | 27.23% |
| | 0.46 | 96.2 | 308.48 | 477.95 | 23.28% |
| | 0.48 | 96.0 | 279.91 | 444.97 | 21.12% |
| | 0.49 | 96.1 | 278.02 | 447.05 | 20.98% |
| | 0.495 | 96.2 | 275.14 | 444.18 | 20.76% |
| | 0.498 | 96.1 | 272.52 | 437.76 | 20.57% |
| | 0.5 | 95.9 | 110.27 | 284.48 | 8.32% |
| | 0.6 | 95.6 | 105.16 | 285.95 | 7.94% |
| | 0.7 | 95.4 | 104.50 | 287.29 | 7.89% |
| | 0.8 | 95.4 | 102.56 | 291.24 | 7.74% |
| | 0.9 | 94.9 | 102.59 | 288.54 | 7.74% |
| | 1.0 (Distilled) | 94.7 | 102.45 | 297.50 | 7.73% |
| | 0 (Qwen3-14B) | 95.4 | 4109.18 | 4865.18 | 100.00% |
| | 0.1 | 95.4 | 3924.72 | 4629.28 | 95.51% |
| | 0.2 | 95.6 | 3572.30 | 4229.31 | 86.93% |
| | 0.25 | 95.2 | 3347.99 | 3895.70 | 81.48% |
| | 0.275 | 94.9 | 3089.26 | 3593.51 | 75.18% |
| | 0.3 | 95.0 | 2978.05 | 3488.89 | 72.47% |
| | 0.325 | 94.5 | 2813.51 | 3299.48 | 68.47% |
| | 0.35 | 94.6 | 2756.00 | 3246.53 | 67.07% |
| | 0.375 | 93.9 | 2500.21 | 2972.72 | 60.84% |
| | 0.4 | 92.6 | 2227.13 | 2697.19 | 54.20% |
| | 0.42 | 91.9 | 2047.34 | 2508.61 | 49.82% |
| MATH | 0.44 | 91.3 | 1911.97 | 2373.67 | 46.53% |
| | 0.46 | 90.5 | 1763.34 | 2223.73 | 42.91% |
| | 0.48 | 90.3 | 1697.39 | 2168.32 | 41.31% |
| | 0.49 | 90.0 | 1675.61 | 2163.59 | 40.78% |
| | 0.495 | 90.2 | 1677.61 | 2121.31 | 40.83% |
| | 0.498 | 89.1 | 1660.45 | 2125.71 | 40.41% |
| | 0.5 | 79.8 | 421.82 | 888.41 | 10.27% |
| | 0.6 | 77.1 | 303.95 | 775.21 | 7.40% |
| | 0.7 | 76.3 | 284.95 | 736.22 | 6.93% |
| | 0.8 | 73.4 | 261.47 | 733.30 | 6.36% |
| | 0.9 | 71.6 | 247.82 | 713.24 | 6.03% |
| | 1.0 (Distilled) | 70.5 | 244.44 | 686.38 | 5.95% |

about 15,000 samples from the GSM8K and MATH mixture dataset, using the `qwen3` prompt template. A cutoff length of 32,768 tokens is applied to all sequences.

The model is trained for 3 epochs using the `adamw_torch` optimizer and the following hyperparameters:

- **Cutoff length:** 32,768

- **Max samples:** 15,000

- **Batch size:** 1 (with gradient accumulation of 8)

- **Learning rate:** $2 \times 10^{-5}$ with a cosine schedule and a warmup ratio of 0.1

- **Precision:** `bf16`

- **LoRA rank:** 256

- **LoRA alpha:** 16

- **Validation split:** 10% of the training data

- **Evaluation strategy:** every 200 steps

The training is accelerated with performance optimizations, including the Liger kernel and Unsloth's garbage collector. All experiments utilize 16 parallel workers for data preprocessing and are configured with `overwrite_cache=true`.

### B.5 DETAILS OF EVALUATION

We use the Qwen2.5-Math (Yang et al., 2024) framework for unified evaluation across tasks. In our evaluation setup, all models were constrained to a maximum generation length of 32,768 tokens and temperature of 0.6 to align with DeepSeek' technical report (Guo et al., 2025) and Qwen3' technical report (Yang et al., 2025).

## C DISCUSSIONS

### C.1 PROMPT-BASED LENGTH CONTROL

To evaluate the precision of prompt-based methods in controlling reasoning chain length, we conduct experiments using explicit length control prompts with the Qwen3-8B model on the MATH-500 dataset. We employ a structured prompt template that instructs the model to reduce the word count in its Chain-of-Thought process by specified percentages (ranging from 10% to 90% reduction).

The following example shows the prompt template used for 90% word reduction:

```
Please reduce 90% of the tokens in your Chain-of-Thought process.
```

For different compression ratios, we adjust the percentage value in the system prompt accordingly.

As shown in Table 5, the results demonstrate that prompt-based methods cannot precisely control the compression ratio. For instance, when instructing the model to reduce words by 90%, the actual ACT reduction is only 42.7%. Similarly, for 50% word reduction, the ACT reduction is 37.2%. This inconsistency indicates that prompt-based length control is imprecise and cannot reliably achieve the desired compression levels, highlighting the limitation of relying solely on prompt engineering for fine-grained reasoning length adjustment.

Table 5: Performance of prompt-based length control on MATH-500 dataset (Qwen3-8B). Length constraints are expressed as percentages of the original reasoning chain length. Compression ratios (in red subscript) indicate ACT reduction compared to vanilla baseline.

| Compression Ratio | Metrics | | |
| --- | --- | --- | --- |
| in Prompt | Pass@1 | ACT | AAT |
| Vanilla | 94.6 | $4602.15_{-0.0\%}$ | 5419.64 |
| 10% | 94.0 | $3280.58_{-28.7\%}$ | 3930.71 |
| 20% | 95.4 | $3355.98_{-27.1\%}$ | 3935.61 |
| 30% | 94.4 | $3204.56_{-30.4\%}$ | 3824.41 |
| 40% | 95.4 | $3241.77_{-29.6\%}$ | 3808.46 |
| 50% | 95.8 | $2891.57_{-37.2\%}$ | 3508.62 |
| 60% | 95.2 | $3145.84_{-31.6\%}$ | 3748.60 |
| 70% | 94.6 | $3009.26_{-34.6\%}$ | 3545.86 |
| 80% | 95.6 | $2782.41_{-39.5\%}$ | 3355.50 |
| 90% | 95.2 | $2636.84_{-42.7\%}$ | 3208.47 |

## C.2 EFFECTIVENESS OF MODEL FUSION ACROSS ARCHITECTURES

To validate the general effectiveness of model fusion approach across different model architectures, we conduct comprehensive experiments with both Qwen3-8B and DeepSeek-R1-Distill-Qwen-7B models. Consistent with the methodology described in Section 4.5.1, we sample 11 values of $\alpha \in \{0, 0.1, ..., 0.9, 1\}$ to create discrete ensembles for both model families.

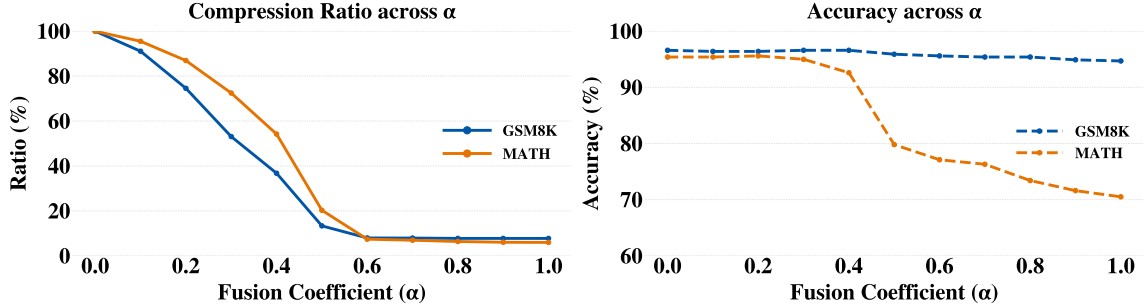

Figure 7: Effect of fusion coefficient ($\alpha$) on compression ratio and accuracy for Qwen3-8B.

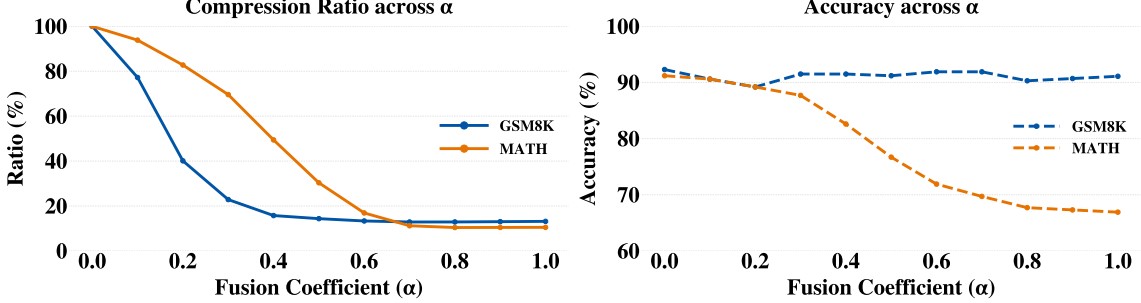

Figure 8: Effect of fusion coefficient ($\alpha$) on compression ratio and accuracy for DeepSeek-R1-Distill-Qwen-7B.

The results, illustrated in Figure 7 and Figure 8, demonstrate that model fusion consistently creates smooth continua of reasoning styles regardless of the base model architecture, confirming the key observations from our main analysis:

**Consistent Compression Behavior.** Both models exhibit the characteristic smooth, monotonically decreasing relationship between $\alpha$ and reasoning length. The compression ratio progresses predictably from 0% to over 90% as $\alpha$ increases from 0 to 1, providing fine-grained control over reasoning complexity without additional training.

**Robust Accuracy Patterns.** The nuanced accuracy relationship observed in our main experiments is replicated across both architectures. As $\alpha$ increases from 0 to 1, accuracy initially shows slight improvement before entering a declining phase, reaffirming that longer reasoning chains do not invariably lead to higher accuracy and excessive verbosity can be counterproductive.

**Architectural Independence.** The consistent fusion behavior across both the Qwen3 and DeepSeek model families demonstrates that our parameter interpolation method is not architecture-specific. Both models successfully maintain the efficiency-accuracy Pareto frontier sampling capability, enabling optimal point selection for adaptive training.

These cross-architecture results validate model fusion as a robust and generalizable technique for generating reasoning spectra, providing the foundation for DART's adaptive reasoning capabilities. The consistent performance patterns observed across both model families underscore the reliability of our approach for creating efficient adaptive reasoning systems in diverse deployment scenarios.

# D  CASE STUDY

## D.1  CASE STUDY: SHORT COT DATA DISTILLATION

This case study in Table 6 demonstrates the effectiveness of our short Chain-of-Thought (CoT) data distillation process using the DeepSeek-R1 model guided by the prompt template shown in Figure 6. The example illustrates how a verbose reasoning chain is condensed while preserving logical integrity and key computational steps.

The distillation process successfully adheres to all prompt requirements:

**Preserves original reasoning logic.** The distilled CoT maintains the core mathematical approach of setting up the equation $\frac{8}{7}x = 240$ and solving for $x$.

**Removes redundant descriptions.** Extensive explanatory text and self-reflection statements are eliminated, reducing the reasoning chain by approximately 70% in length.

**Ensures independent task completion.** The condensed reasoning contains all necessary steps to solve the problem independently.

**Achieves maximum conciseness.** The output presents only essential mathematical operations and logical transitions.

The distilled CoT demonstrates efficient knowledge transfer while maintaining mathematical accuracy, validating our approach for generating compact yet faithful reasoning chains. This case exemplifies how DART's distillation step creates the concise reasoning patterns necessary for building the adaptive reasoning spectrum.

| **Case Study: Short CoT Data Distillation** |
|---|
| **Input Question:** |
| The school has 240 popular science books, which is more than story books $\frac{1}{7}$. How many story books are there? |
| **Original Long CoT:** |
| Well, OK, I have to solve this problem now. The question is that the school has 240 popular science books, which is $\frac{1}{7}$ more than the story books. How many story books are there? I need to carefully analyze the relationship between the various quantities in the question and then find the correct solution. |

First of all, the question mentions that there are 240 popular science books, and popular science books are $\frac{1}{7}$ more than story books. The "$\frac{1}{7}$ more than story books" here is the key. I need to make it clear that the comparison object here is the number of story books, so the number of popular science books is the number of story books plus $\frac{1}{7}$ of story books, or $(1 + \frac{1}{7})$ times the number of story books. Suppose the number of story books is $x$. Then according to the description of the question, the number of popular science books is equal to the number of story books plus $\frac{1}{7}$ of it, that is, $x + \frac{1}{7}x$, or it can be written as $(1 + \frac{1}{7})x$, which is $\frac{8}{7}x$. The question already gives the number of popular science books as 240, so we can build an equation: $\frac{8}{7}x = 240$.

Next, I need to solve this equation to find the value of $x$. To find $x$, I can multiply both sides by $\frac{7}{8}$, so that the coefficient on the left can be canceled. To calculate specifically: $x = 240 \times \frac{7}{8}$.

Then, I need to calculate this multiplication. First, divide 240 by 8, because the denominator is 8, which makes the calculation easier. 240 divided by 8 equals 30, and then multiplied by 7, we get $30 \times 7 = 210$. So $x$ equals 210.

However, I need to double-check whether this process is correct and whether there is any mistake in the unit or proportion. For example, the question says that the number of popular science books is $\frac{1}{7}$ more than the story books. The $\frac{1}{7}$ here is $\frac{1}{7}$ of the story books, not the number of popular science books. So the correct expression should be that the number of popular science books is equal to the number of story books plus $\frac{1}{7}$ of the number of story books, which is $\frac{8}{7}$ times the number of story books. If we substitute the numerical value, we should get 210 story books. Popular science books are $\frac{8}{7}$ of 210, which is $210 \times \frac{8}{7} = 240$, which just meets the conditions given in the question. So my calculation should be correct.

In summary, there are $\boxed{210}$ story books.

| **Distilled Short CoT:** |
|---|

The school has 240 popular science books, which is $\frac{1}{7}$ more than the story books. Let the number of story books be $x$. The popular science books are then $x + \frac{1}{7}x = \frac{8}{7}x$. Given $\frac{8}{7}x = 240$, solving for $x$ yields $x = 240 \times \frac{7}{8} = 210$. Verification: $\frac{8}{7} \times 210 = 240$, confirming the answer. Thus, there are $\boxed{210}$ story books.

| **Original Answer:** |
|---|

According to the question, there are $\frac{1}{7}$ more popular science books than story books, that is, the number of popular science books is $1 + \frac{1}{7} = \frac{8}{7}$ times that of story books. Assume there are $x$ story books, then the equation is: $\frac{8}{7}x = 240$. Solved: $x = 240 \times \frac{7}{8} = 210$. **Verification**: $\frac{8}{7}$ times 210 story books is $210 \times \frac{8}{7} = 240$, which is consistent with the question conditions. Therefore, there are $\boxed{210}$ story books.

Table 6: Case Study: Short CoT Data Distillation

