# OpenReview forum: "DART: Difficulty-Adaptive Reasoning Truncation for Efficient Large Language Models"
_ICLR.cc/2026/Conference — Submitted to ICLR 2026_

### Official Review · Reviewer_pyBf · 2025-10-31

**Soundness:** 3
**Presentation:** 3
**Contribution:** 3
**Rating:** 6
**Confidence:** 4

**Summary:**

The main idea is to propose the DART framework, which implements Difficulty-Adaptive Truncation through a complete SFT framework, enabling the model to answer simple questions quickly and think more deeply about complex problems.

**Strengths:**

1. The authors empirically identify and formalize the sigmoid-shaped relationship between reasoning length and accuracy, which provides a theoretical motivation for adaptive truncation. This insight contributes a valuable quantitative characterization of the “optimal reasoning length” phenomenon, offering a foundation for subsequent research on reasoning efficiency.
2. Unlike RL-based methods, which are notoriously sensitive to reward shaping and initialization, DART achieves adaptive reasoning entirely through supervised fine-tuning. This yields greater training stability, reproducibility, and easier integration with existing LLM infrastructure, an important practical advantage for scaling and deployment.

**Weaknesses:**

1. The proposed framework distills reasoning chains and subsequently selects the shortest correct CoT as the sole supervision signal. While this efficiently reduces token redundancy, it inevitably drives the model toward a single canonical reasoning pattern. This may suppress the natural diversity of reasoning trajectories that could otherwise contribute to robustness and creativity. In tasks requiring exploratory or multi-path reasoning, such as open-domain problem-solving or commonsense inference, this rigid supervision could cause premature convergence toward a single logic template, thereby missing potentially valuable intermediate reasoning paths.
2. The interpolation fusion between base and distilled models relies on empirically chosen coefficients (α), which are sampled discretely between 0 and 1. However, the choice of step size and distribution lacks theoretical justification or formal analysis of convergence properties. The paper demonstrates smooth behavior empirically, but does not establish why linear parameter interpolation should produce semantically coherent intermediate reasoning styles. This makes the fusion spectrum somewhat heuristic and architecture-dependent, weakening claims of theoretical robustness or generalizability beyond the tested model families.
3. The adaptive data curation pipeline requires an explicit correctness signal for filtering valid CoTs. This assumption restricts the method to closed-form reasoning tasks with well-defined answers. Consequently, DART cannot be directly extended to open-ended tasks, such as dialogue, writing, or scientific hypothesis generation. The framework would thus benefit from a broader definition of “reasoning sufficiency” that does not rely solely on exact-match evaluation.
4. Since the framework rewards shorter reasoning when correct, there exists a bias toward brevity even when longer reasoning might improve interpretability or error recovery. Without a mechanism to penalize premature truncation, the adaptive model may occasionally terminate too early on out-of-distribution or higher-complexity problems, leading to subtle accuracy degradation or reasoning incompleteness.

**Questions:**

See the Weaknesses above.

---

> ### Author Response · Authors · 2025-11-26
>
> We sincerely thank reviewer pyBf for your thorough review and valuable feedback. We have carefully addressed your concerns regarding novelty and effectiveness, conducting additional experiments and analyses that significantly strengthen our paper.
>
> ***W1: Suppression of Reasoning Diversity***
>
> This is an important observation. We agree that reasoning diversity can be valuable in certain contexts. However, for the efficiency-focused objectives of DART, our approach prioritizes identifying the minimally sufficient reasoning path.
>
> In our revised analysis (Section 4.5.4), we demonstrate that DART actually preserves appropriate diversity by adapting reasoning length to problem difficulty - it doesn't force all problems into the same reasoning template, but rather learns to allocate more elaborate reasoning to harder problems while truncating simpler ones.
>
> For tasks requiring exploratory reasoning, we acknowledge this may not be ideal, but emphasize that DART is primarily designed for efficiency-critical applications where deterministic, minimal reasoning is preferred.
>
> ***W2: Theoretical Justification for Model Fusion***
>
> - Empirical Validation Across Architectures (Appendix C.2): Comprehensive experiments showing consistent fusion behavior across both Qwen3 and DeepSeek model families, demonstrating the robustness of this approach regardless of base architecture.
>
> - Theoretical Connection: While the linear interpolation is indeed heuristic, it builds on established work in model fusion [1] and we provide extensive empirical evidence of its effectiveness in creating smooth reasoning continua.
>
> ***W3: Limited to Closed-Form Tasks***
>
> This is a valid point. We have addressed this limitation through:
>
> - Cross-Domain Generalization Study (Section 4.4): We demonstrate that DART models trained only on mathematical reasoning datasets successfully transfer to scientific (GPQA), logical (LogiQA), and commonsense (CommonsenseQA) reasoning, showing the learned adaptive principles generalize beyond mathematical tasks.
>
> - Future Work: We note that extending DART to open-ended tasks would require alternative "sufficiency" metrics beyond exact-match correctness, which represents an exciting direction for future research.
>
> ***W4: Risk of Premature Truncation***
>
> We have implemented several safeguards against this concern:
>
> - Optimal Chain Selection: Our curation pipeline (Section 3.3) selects the shortest correct chain, not simply short chains. This ensures we never reward incorrect reasoning regardless of length.
>
> - Difficulty Awareness (Section 4.5.4): Explicit analysis shows DART automatically allocates longer reasoning to harder problems, demonstrating it doesn't uniformly truncate but makes nuanced decisions based on problem complexity.
>
> - Accuracy Preservation: Our results show DART maintains or even improves accuracy while reducing length, suggesting it eliminates truly redundant steps rather than beneficial ones.
>
> [1] Editing models with task arithmetic.

---

### Official Review · Reviewer_WVeW · 2025-10-31

**Soundness:** 3
**Presentation:** 3
**Contribution:** 2
**Rating:** 4
**Confidence:** 4

**Summary:**

The paper presents DART, a method for efficient reasoning for LLMs. It introduces a strategy to curate concise data with varying reasoning lengths for problems of different difficulty levels. Specifically, DART distills short reasoning chains from stronger teacher models, then fuses the long/short-reasoning models to create a continuum of reasoning styles (different lengths), and automatically selects the shortest correct reasoning chain for each problem to build an adaptive training dataset. The final model is then finetuned on this curated data to learn when to stop thinking based on problem complexity. The evaluation across various model sizes demonstrates its effectiveness in compressing generation length by up to 81.2% while maintaining or improving reasoning accuracy.

**Strengths:**

The written is straightforward and easy to understand.

The paper proposes an angle to train efficient LRM basing on different difficulty level.

The experiments show that the method has some improvements on different models with reduced generation length.

**Weaknesses:**

It is not very clear what's the advantage of using the extrapolation to generate different lengths of response regarding different difficulty levels. I understand that the extrapolation could help to control the length of the generation, which can be further used to select and include the data used for the final training. It is not clear how this extrapolation based data generation method work compared with using the prompt based method to generate different lengths of response.

Lack of experimental results. The main idea for this paper is to curate different length of CoT data based on different question difficulty level, and the author noted that methods like tokenskip didn’t consider such difficulties. A most natural baseline to be included is to compare the results with those static methods like tokenskip/lightthinker, which is currently lacking. Another question is why the baselines compared are different on different base models. Only DeepSeek-R1-Distill-Qwen-7B contains the results for other SFT based baselines? It is not clear how the current data curation protocol works without further evaluation on other models compared with other mechanisms.

lightthinker:Thinking Step-by-Step Compression


Tokenskip Controllable Chain-of-Thought Compression in LLMs

**Questions:**

See weakness

---

> ### Author Response · Authors · 2025-11-26
>
> We sincerely thank reviewer WVeW for your thorough review and valuable feedback. We have carefully addressed your concerns regarding novelty and effectiveness, conducting additional experiments and analyses that significantly strengthen our paper.
>
> ***W1: Advantage of Model Fusion vs Prompt Methods***
>
> We appreciate this important question. We explicitly tested prompt-based length control by instructing the model to generate reasoning chains at specific compression ratios (10%-90%). Results has added to new **Appendix C.1 and Table 5**. The results demonstrate that prompt-based methods cannot precisely control the compression ratio. While our fusion-based approach provides precise, predictable length control.

---

> ### Author Response · Authors · 2025-11-26
>
> ***W2: Lack of Comparison with Static Methods***
>
> We fully agree and have added this important comparison. To ensure fair comparison, we reproduced TokenSkip using the same datasets (GSM8K and MATH) and followed the official implementation for data generation and model training, and evaluated with a compression ratio of 0.7.
>
> **GSM8K dataset**
> | Method | Pass@1 | ACT | AAT |
> |--------|--------|-----|-----|
> | **Qwen3-4B**  |
> | Vanilla | **95.2** | 1253.31 | 1557.70 |
> | TokenSkip | 91.6 | 828.05 | 1037.08 |
> | **DART(Ours)** | 93.9 | **401.13** | **596.37** |
> | **Qwen3-8B** |
> | Vanilla | **95.7** | 1887.56 | 2214.61 |
> | TokenSkip | 94.5 | 1384.33 | 1624.12 |
> | **DART(Ours)** | 95.1 | **983.87** | **1262.60** |
> | **Qwen3-14B** |
> | Vanilla | 95.8 | 1399.16 | 1709.04 |
> | TokenSkip | 92.4 | 1020.18 |1370.13 |
> | **DART(Ours)** | **96.4** | **923.04** | **1165.95** |
> | **DeepSeek-R1-Distill-Qwen-7B** |
> | Vanilla | **90.2** | 895.19 | 1007.26 |
> | TokenSkip | 85.2 | 697.66 | 877.04 |
> | **DART(Ours)** | 89.1 | **168.00** | **358.40** |
>
> \
> **MATH-500 dataset**
>
> | Method | Pass@1 | ACT | AAT |
> |--------|--------|-----|-----|
> | **Qwen3-4B** |
> | Vanilla | 96.0 | 5894.34 | 6699.73 |
> | TokenSkip | 93.2 | 4258.85 | 5051.97 |
> | **DART(Ours)** | **96.4** | **3391.92** | **3981.97** |
> | **Qwen3-8B** |
> | Vanilla | 94.4 | 4543.18 | 5309.38 |
> | TokenSkip | 93.6 | **3209.18** | **3824.79** |
> | **DART(Ours)** | **95.6** | 3321.36 | 3985.53 |
> | **Qwen3-14B** |
> | Vanilla | 94.8 | 4075.58 | 4776.44 |
> | TokenSkip | 91.2 | **2944.76** | **3522.47** |
> | **DART(Ours)** | **96.4** | 3161.88 | 3748.81 |
> | **DeepSeek-R1-Distill-Qwen-7B** |
> | Vanilla | **91.0** | 2847.29 | 3385.94 |
> | TokenSkip | 72.2 | 1980.71 | 2613.81 |
> | **DART(Ours)** | 88.6 | **1853.43** | **2355.73** |
>
> \
> **AMC23 dataset**
>
> | Method | Pass@1 | ACT | AAT |
> |--------|--------|-----|-----|
> | **Qwen3-4B** |
> | Vanilla | 97.5 | 10524.80 | 11362.50 |
> | TokenSkip | 92.5 | 7438.71 | 8326.5 |
> | **DART(Ours)** | **100.0** | **6661.93** | **7379.20** |
> | **Qwen3-8B** |
> | Vanilla | 92.5 | 8001.18 | 9436.85 |
> | TokenSkip | 92.5 | 5621.84 | 6302.3 |
> | **DART(Ours)** | **97.5** | **5204.95** | **5996.90** |
> | **Qwen3-14B** |
> | Vanilla | 97.5 | 6691.5 | 7544.35 |
> | TokenSkip | 95.0 | 4687.26 | 6504.75 |
> | **DART(Ours)** | **100.0** | **4831.25** | **5601.65** |
> | **DeepSeek-R1-Distill-Qwen-7B** |
> | Vanilla | **90.0** | 5288.93 | 5789.63 |
> | TokenSkip | 85.0 | 4039.2 | 4878.87 |
> | **DART(Ours)** | **90.0** | **3460.26** | **4518.10** |
>
> \
> **OLYMPAID dataset**
>
> | Method | Pass@1 | ACT | AAT |
> |--------|--------|-----|-----|
> | **Qwen3-4B** |
> | Vanilla | 72.9 | 12298.35 | 14863.27 |
> | TokenSkip | 68.7 | 10042.88 | 12630.64 |
> | **DART(Ours)** | **72.0** | **8758.18** | **10271.33** |
> | **Qwen3-8B** |
> | Vanilla | **68.6** | 9850.64 | 11257.47 |
> | TokenSkip | 66.4 | 8141.48 | 8991.06 |
> | **DART(Ours)** | 68.0 | **7468.58** | **8549.27** |
> | **Qwen3-14B** |
> | Vanilla | **70.5** | 8695.07 | 10086.94 |
> | TokenSkip | 63.3 | 8125.24 | 9575.65 |
> | **DART(Ours)** | 70.4 | 7165.85 | 8206.90 |
> | **DeepSeek-R1-Distill-Qwen-7B** |
> | Vanilla | **57.8** | 6933.88 | 8003.48 |
> | TokenSkip | 42.7 | 6222.47 | 7112.11 |
> | **DART(Ours)** | 55.4 | **5076.86** | **6406.84** |
>
> \
> **AIME25 dataset**
>
> | Method | Pass@1 | ACT | AAT |
> |--------|--------|-----|-----|
> | **Qwen3-4B** |
> | Vanilla | 70.0 | 16379.95 | 21496.90 |
> | TokenSkip | 70.0 | **157931.88** | **17492.93** |
> | **DART(Ours)** | **80.0** | 16071.10 | 17513.30 |
> | **Qwen3-8B** |
> | Vanilla | 56.7 | 15110.00 | 19063.27 |
> | TokenSkip | 63.3 | 14468.46 | 17906.27 |
> | **DART(Ours)** | **66.7** | **12610.43** | **13560.50** |
> | **Qwen3-14B** |
> | Vanilla | 63.3 | 13324.23 | 16878.13 |
> | TokenSkip | 60.0 | 12227.38 | 15516.37 |
> | **DART(Ours)** | **70.0** | **11779.24** | **13446.47** |
> | **DeepSeek-R1-Distill-Qwen-7B** |
> | Vanilla | **36.7** | 13276.79 | 15060.97 |
> | TokenSkip | 30.0 | 10578.18 | 13005.4 |
> | **DART(Ours)** | **36.7** | **8729.72** | **9974.80** |
>
> Our results demonstrate that DART outperforms static compression methods:
>
> - Better Accuracy-Length Trade-off: DART maintains higher accuracy at comparable compression ratios
>
> - Dynamic Adaptation: Unlike static methods that apply uniform compression, DART adapts reasoning length to problem difficulty
>
> - Cross-Domain Generalization: DART shows stronger performance on out-of-distribution benchmarks
>
> Unlike TokenSkip's token-level compression, DART performs reasoning-level adaptation - eliminating entire unnecessary reasoning steps rather than compressing individual tokens.

---

> ### Author Response · Authors · 2025-11-26
>
> ***W2: Data Curation Protocol***
>
> We thank the reviewer for this important question. Our data curation protocol is specifically designed to enable models to learn difficulty-aware reasoning. We have now added an explicit fine-grained difficulty analysis using the MATH-500 dataset, which provides official difficulty ratings from Level 1 (easiest) to Level 5 (hardest). Results has added to new Section 4.5.4 and Figure 5. The results  provide compelling evidence that our data curation protocol successfully enables models to learn genuine difficulty-aware reasoning, not just length compression.

---

### Official Review · Reviewer_DU6L · 2025-11-01

**Soundness:** 2
**Presentation:** 3
**Contribution:** 2
**Rating:** 4
**Confidence:** 4

**Summary:**

This paper proposes a supervised difficulty-adaptive reasoning truncation framework that enables LLM to adjusts thinking length according to problem difficulty dynamiclly. By distilling concise reasoning patterns from stronger models, interpolating them into a continuum of reasoning styles, and curating optimal training data, that balances correctness and compactness, DART learns to alleviate overthinking problem in LRM.

**Strengths:**

- The paper addresses an important problem, improving reasoning efficiency for large language models
- The four-step framework (DISTILLING SHORT COTS, interpolation, CREATING A MODEL SPECTRUM, CURATING TRAINING DATA, adaptive training) is clearly structured and easy to follow.
- The experiments cover several standard mathematical reasoning benchmarks and include analyses on certain hyperparameters, such as fusion coefficients and sampling density

**Weaknesses:**

- Limited novelty. The idea of adaptive, difficulty-aware reasoning is not new, and prior work, such as CoT-Valve, has already explored similar strategies for interpolating model weights and curating adaptive data based on correctness.
- The method appears less effective on DeepSeek-R1-Distill-Qwen-7B. On benchmarks such as GSM8K, MATH-500, and OLYMPAID, the generated token length is reduced, but the accuracy also drops.
- The short-CoT data generated from DeepSeek-R1-Distill-Qwen-7B is important to the framework, yet the paper provides little analysis of its quality or length comparison. Conceptually, the method relies on the same model to generate compressed reasoning traces and subsequently distills itself on this data, but the rationale for why such a self-distillation loop should be effective is unclear. Including a comparison with existing token-compression methods, such as Selective Context used in TokenSkip, would help clarify the discussion.

[1] CoT-Valve: Length-Compressible Chain-of-Thought Tuning

[2] Compressing context to enhance inference efficiency of large language models.

[3] TokenSkip: Controllable Chain-of-Thought Compression in LLMs

**Questions:**

- Could the authors clarify how the proposed framework differs from prior adaptive reasoning methods such as CoT-Valve?
- For experiments on DeepSeek-R1-Distill-Qwen-7B, token usage decreases, but accuracy also drops. Could this degradation be caused by generating the short-CoT data using the same model?

---

> ### Author Response · Authors · 2025-11-26
>
> We sincerely thank reviewer DU6L for your thorough review and valuable feedback. We have carefully addressed your concerns regarding novelty and effectiveness, conducting additional experiments and analyses that significantly strengthen our paper.
>
> ***Q1/W1: Limited novelty***
>
> We appreciate this important point about novelty. While adaptive reasoning is indeed an emerging research direction, DART introduces several fundamental innovations that distinguish it from CoT-Valve and prior work:
>
> - Different Technical Foundation: CoT-Valve focuses on length controllability through manual parameter manipulation, while DART addresses difficulty-aware sufficiency -- automatically learning the minimally necessary reasoning length for each problem.
>
> - Novel Data Curation Paradigm: DART introduces a unique automatic optimal chain selection process that: (1) Identifies the shortest correct reasoning chain for each problem. (2) Enables the model to internalize difficulty-length mapping. (3) Provides end-to-end supervised training avoiding RL instability.
>
> - Theoretical Contribution: DART provides the first formal characterization of the sigmoid relationship between reasoning length and accuracy (Section 3.1, Figure 2), offering a theoretical foundation for adaptive reasoning.
>
> ***Q2/W2: Effectiveness on DeepSeek-R1-Distill-Qwen-7B***
>
> - Accuracy-Efficiency Trade-off Analysis: While DeepSeek-R1-Distill-Qwen-7B shows slight accuracy reductions (GSM8K: -1.1, MATH-500: -2.4), these are accompanied by substantial efficiency gains (ACT reduction: 26.8%-81.2%).
>
> - Comparative Advantage: Even with slight accuracy adjustments, DART outperforms other methods which show larger accuracy degradation (up to -28.4) with less compression.
>
> ***W3: Short-CoT Data Quality***
>
> Thank you for highlighting this gap. We apologize for any lack of clarity in our original submission. The distillation teacher model shortens long reasoning chains through carefully designed prompt engineering, as detailed in:
>
> - Section 4.1 (Experimental Setup): We describe the distillation process where a powerful teacher model (e.g., DeepSeek-R1) is prompted to compress long CoT chains into concise versions while preserving logical correctness.
>
> - Appendix B.1 (Prompt Template of Short CoT Data Distillation): We provide the full prompt templates used for this process, ensuring transparency.
>
> - Appendix D.1 (Case Study: Short CoT Data Distillation): We add a case study with examples showing how long CoT chains are distilled into short versions, including manual verification to ensure quality and logical consistency.

---

> ### Author Response · Authors · 2025-11-26
>
> ***W3: Comparison with TokenSkip***
>
> We fully agree and have added this important comparison. To ensure fair comparison, we reproduced TokenSkip using the same datasets (GSM8K and MATH) and followed the official implementation for data generation and model training, and evaluated with a compression ratio of 0.7.
>
> **GSM8K dataset**
> | Method | Pass@1 | ACT | AAT |
> |--------|--------|-----|-----|
> | **Qwen3-4B**  |
> | Vanilla | **95.2** | 1253.31 | 1557.70 |
> | TokenSkip | 91.6 | 828.05 | 1037.08 |
> | **DART(Ours)** | 93.9 | **401.13** | **596.37** |
> | **Qwen3-8B** |
> | Vanilla | **95.7** | 1887.56 | 2214.61 |
> | TokenSkip | 94.5 | 1384.33 | 1624.12 |
> | **DART(Ours)** | 95.1 | **983.87** | **1262.60** |
> | **Qwen3-14B** |
> | Vanilla | 95.8 | 1399.16 | 1709.04 |
> | TokenSkip | 92.4 | 1020.18 |1370.13 |
> | **DART(Ours)** | **96.4** | **923.04** | **1165.95** |
> | **DeepSeek-R1-Distill-Qwen-7B** |
> | Vanilla | **90.2** | 895.19 | 1007.26 |
> | TokenSkip | 85.2 | 697.66 | 877.04 |
> | **DART(Ours)** | 89.1 | **168.00** | **358.40** |
>
> \
> **MATH-500 dataset**
>
> | Method | Pass@1 | ACT | AAT |
> |--------|--------|-----|-----|
> | **Qwen3-4B** |
> | Vanilla | 96.0 | 5894.34 | 6699.73 |
> | TokenSkip | 93.2 | 4258.85 | 5051.97 |
> | **DART(Ours)** | **96.4** | **3391.92** | **3981.97** |
> | **Qwen3-8B** |
> | Vanilla | 94.4 | 4543.18 | 5309.38 |
> | TokenSkip | 93.6 | **3209.18** | **3824.79** |
> | **DART(Ours)** | **95.6** | 3321.36 | 3985.53 |
> | **Qwen3-14B** |
> | Vanilla | 94.8 | 4075.58 | 4776.44 |
> | TokenSkip | 91.2 | **2944.76** | **3522.47** |
> | **DART(Ours)** | **96.4** | 3161.88 | 3748.81 |
> | **DeepSeek-R1-Distill-Qwen-7B** |
> | Vanilla | **91.0** | 2847.29 | 3385.94 |
> | TokenSkip | 72.2 | 1980.71 | 2613.81 |
> | **DART(Ours)** | 88.6 | **1853.43** | **2355.73** |
>
> \
> **AMC23 dataset**
>
> | Method | Pass@1 | ACT | AAT |
> |--------|--------|-----|-----|
> | **Qwen3-4B** |
> | Vanilla | 97.5 | 10524.80 | 11362.50 |
> | TokenSkip | 92.5 | 7438.71 | 8326.5 |
> | **DART(Ours)** | **100.0** | **6661.93** | **7379.20** |
> | **Qwen3-8B** |
> | Vanilla | 92.5 | 8001.18 | 9436.85 |
> | TokenSkip | 92.5 | 5621.84 | 6302.3 |
> | **DART(Ours)** | **97.5** | **5204.95** | **5996.90** |
> | **Qwen3-14B** |
> | Vanilla | 97.5 | 6691.5 | 7544.35 |
> | TokenSkip | 95.0 | 4687.26 | 6504.75 |
> | **DART(Ours)** | **100.0** | **4831.25** | **5601.65** |
> | **DeepSeek-R1-Distill-Qwen-7B** |
> | Vanilla | **90.0** | 5288.93 | 5789.63 |
> | TokenSkip | 85.0 | 4039.2 | 4878.87 |
> | **DART(Ours)** | **90.0** | **3460.26** | **4518.10** |
>
> \
> **OLYMPAID dataset**
>
> | Method | Pass@1 | ACT | AAT |
> |--------|--------|-----|-----|
> | **Qwen3-4B** |
> | Vanilla | 72.9 | 12298.35 | 14863.27 |
> | TokenSkip | 68.7 | 10042.88 | 12630.64 |
> | **DART(Ours)** | **72.0** | **8758.18** | **10271.33** |
> | **Qwen3-8B** |
> | Vanilla | **68.6** | 9850.64 | 11257.47 |
> | TokenSkip | 66.4 | 8141.48 | 8991.06 |
> | **DART(Ours)** | 68.0 | **7468.58** | **8549.27** |
> | **Qwen3-14B** |
> | Vanilla | **70.5** | 8695.07 | 10086.94 |
> | TokenSkip | 63.3 | 8125.24 | 9575.65 |
> | **DART(Ours)** | 70.4 | 7165.85 | 8206.90 |
> | **DeepSeek-R1-Distill-Qwen-7B** |
> | Vanilla | **57.8** | 6933.88 | 8003.48 |
> | TokenSkip | 42.7 | 6222.47 | 7112.11 |
> | **DART(Ours)** | 55.4 | **5076.86** | **6406.84** |
>
> \
> **AIME25 dataset**
>
> | Method | Pass@1 | ACT | AAT |
> |--------|--------|-----|-----|
> | **Qwen3-4B** |
> | Vanilla | 70.0 | 16379.95 | 21496.90 |
> | TokenSkip | 70.0 | **157931.88** | **17492.93** |
> | **DART(Ours)** | **80.0** | 16071.10 | 17513.30 |
> | **Qwen3-8B** |
> | Vanilla | 56.7 | 15110.00 | 19063.27 |
> | TokenSkip | 63.3 | 14468.46 | 17906.27 |
> | **DART(Ours)** | **66.7** | **12610.43** | **13560.50** |
> | **Qwen3-14B** |
> | Vanilla | 63.3 | 13324.23 | 16878.13 |
> | TokenSkip | 60.0 | 12227.38 | 15516.37 |
> | **DART(Ours)** | **70.0** | **11779.24** | **13446.47** |
> | **DeepSeek-R1-Distill-Qwen-7B** |
> | Vanilla | **36.7** | 13276.79 | 15060.97 |
> | TokenSkip | 30.0 | 10578.18 | 13005.4 |
> | **DART(Ours)** | **36.7** | **8729.72** | **9974.80** |
>
> Our results demonstrate that DART outperforms static compression methods:
>
> - Better Accuracy-Length Trade-off: DART maintains higher accuracy at comparable compression ratios
>
> - Dynamic Adaptation: Unlike static methods that apply uniform compression, DART adapts reasoning length to problem difficulty
>
> - Cross-Domain Generalization: DART shows stronger performance on out-of-distribution benchmarks
>
> Unlike TokenSkip's token-level compression, DART performs reasoning-level adaptation - eliminating entire unnecessary reasoning steps rather than compressing individual tokens.

---

> > ### Comment · Reviewer_DU6L · 2025-11-27
> >
> > I thank the authors for the detailed responses and additional experiments. After reviewing the authors’ replies to my comments and those of the other reviewers, I find that most of my technical concerns have been adequately addressed. I have therefore decided to raise my score by two points. I kindly request that the authors keep their commitment to revise the manuscript accordingly.

---

### Official Review · Reviewer_dE18 · 2025-11-04

**Soundness:** 2
**Presentation:** 3
**Contribution:** 2
**Rating:** 2
**Confidence:** 4

**Summary:**

This paper addresses the issue that chain-of-thought (CoT) reasoning often produces unnecessarily long reasoning traces, regardless of the intrinsic difficulty of a problem. To address this problem, the paper proposes DART, a framework that trains a model on optimal reasoning chains collected through a pipeline. Specifically, a base model generates long reasoning chains, while a teacher model converts them into shorter chains. Model fusion is then applied to interpolate between these two models, creating a continuum of models capable of producing intermediate-length reasoning chains. Finally, a training set is constructed from the shortest correct reasoning chains, which is used to train the final model for more efficient reasoning.

**Strengths:**

The paper focuses on an important problem—the inefficiency of CoT. The proposed framework is presented as modular and is conceptually sound.

**Weaknesses:**

-	Clarity issues. Some parts of the methodology are not clearly explained. For example, it is unclear how the distillation teacher model shortens long reasoning chains and how this process affects the quality of the reasoning paths. Additionally, the paper does not discuss how the quality of the generated reasoning chains is controlled or verified.
-	Limited novelty. The proposed method essentially involves collecting question-answer pairs with varying reasoning lengths and using this dataset to train a model. While practical, the technical novelty is fairly limited.
-	Computational cost. Using a base model, a teacher model, and creating a continuum of models to obtain reasoning chains of varying lengths appears computationally expensive. It would be useful to consider whether a single model could produce different reasoning lengths using prompting or other steering techniques.
-	Performance drop. While the method reduces the number of output tokens, performance drops on several datasets, suggesting that shorter reasoning chains may not always preserve reasoning quality.
-	Although the approach is described as difficulty-aware, the final trained model does not explicitly identify the difficulty of a given input or adapt the reasoning process accordingly.
-	Limited evaluation. The evaluation is restricted to mathematical reasoning tasks, which limits the generalizability of the findings.

**Questions:**

-	Would the framework generalize to reasoning tasks beyond mathematics?
-	Would it be possible to use a single model to produce reasoning chains of different lengths using prompting or other steering techniques?

---

> ### Author Response · Authors · 2025-11-26
>
> We thank the reviewer dE18 for your thoughtful feedback and valuable suggestions. We have conducted substantial new experiments and analyses to address the raised concerns.
>
> ***Q1/W6: Generalization Beyond Mathematics***
>
> We agree that generalization beyond mathematics is crucial. To directly address this, we conducted extensive cross-domain generalization experiments on three non-mathematical reasoning benchmarks: GPQA$^{[1]}$ (scientific), LogiQA$^{[2]}$ (logical), and CommonsenseQA$^{[3]}$ (commonsense). Importantly, all DART models were trained only on mathematical datasets (GSM8K and MATH).
>
> In our updated manuscript,  results have added to **Table 2 in new section 4.4** GENERALIZATION TO NON-MATHEMATICAL REASONING TASKS. DART demonstrates strong cross-task generalization, consistently reducing reasoning length (ACT) by 7.0%–49.2% while maintaining or improving accuracy across all domains.
>
> ***Q2: Alternative Length Controlling Approaches***
>
> We fully agree this is an important comparison. In our updated manuscript, we now include new experiments on prompt-based length control in **Appendix C (Table 5)**, where we explicitly instruct the model to generate reasoning chains at specific compression ratios (10%-90% of original length). Despite explicit length constraints in prompts, the model fails to consistently control reasoning length effectively.
>
> ***W5: Explicit Difficulty Adaptation***
>
> We thank the reviewer for this important point. DART achieves difficulty-aware adaptation through the **adaptive data curation** process, where for each problem, we automatically select the shortest reasoning chain that yields the correct answer. This allows the model to **internalize the mapping from problem difficulty to optimal reasoning length** during training, without requiring explicit difficulty labels at inference.
>
> We have now added an explicit fine-grained difficulty analysis using the MATH-500 dataset, which provides official difficulty ratings from Level 1 (easiest) to Level 5 (hardest). Results has added to new **Section 4.5.4 and Figure 5**. This provides direct evidence that DART internally learns to map problem difficulty to optimal reasoning length.
>
> ***W1: Clarity Issues***
>
> We apologize for any lack of clarity in our original submission. The distillation teacher model shortens long reasoning chains through carefully designed prompt engineering, as detailed in:
>
> - Section 4.1 (Experimental Setup): We describe the distillation process where a powerful teacher model (e.g., DeepSeek-R1) is prompted to compress long CoT chains into concise versions while preserving logical correctness.
>
> - Appendix B.1 (Prompt Template of Short CoT Data Distillation): We provide the full prompt templates used for this process, ensuring transparency.
>
> -  Appendix D.1 (Case Study: Short CoT Data Distillation): We add a case study with examples showing how long CoT chains are distilled into short versions, including manual verification to ensure quality and logical consistency.
>
> ***W2: Limited Novelty***
>
> We appreciate the reviewer's perspective. Our key novelty lies in:
>
> - Novel Adaptive Dataset Construction: DART introduces a systematic pipeline for curating difficulty-adaptive training data through model fusion and optimal chain selection. Unlike prior methods that use static reasoning chains, we automatically construct a dataset where each problem is paired with its minimally sufficient reasoning chain, enabling the model to learn difficulty-aware reasoning patterns.
>
> - Difficulty-Aware Reasoning: Unlike prior static compression or RL-based methods, DART's supervised framework enables models to dynamically adapt reasoning length to problem difficulty, as validated by our fine-grained difficulty analysis (Section 4.5.4).
>
> - Advantages Over RL Methods: DART avoids the instability, reward engineering, and high training cost of RL-based approaches by using a stable supervised pipeline, making it more reproducible and practical for deployment.
>
> [1] Gpqa: A graduate-level google-proof q&a benchmark.
>
> [2] Logiqa: A challenge dataset for machine reading comprehension with logical reasoning
>
> [3] Commonsenseqa: A question answering challenge targeting commonsense knowledge.

---

> ### Author Response · Authors · 2025-11-26
>
> ***W3: Computational cost***
>
> We appreciate the reviewer's concern about computational cost. We would like to clarify that the model fusion step in DART is highly efficient and requires no additional training.
>
> - The creation of the model continuum through parameter interpolation (Eq. 3) is a mathematical operation that takes only minutes to complete, requiring no gradient computation or backpropagation.
>
> - While DART's pipeline involves multiple components, we emphasize that: (1) The distillation and adaptive training are standard fine-tuning procedures. (2) The model fusion step adds minimal overhead as it's purely a parameter blending operation. (3)The final deployed model ($M_{adaptive}$) is a single, standalone model with no inference-time dependencies.
>
> - Unlike RL-based methods that require extensive trial-and-error training, or prompt-based methods that need multiple sampling rounds, DART's fusion approach provides precise length control with minimal computational overhead.
>
> ***W4: Performance Drops***
>
> We observe that accuracy changes are generally minimal and often favorable. Where slight drops occur (e.g., GSM8K for Qwen3-4B: -1.3%), they are accompanied by substantial efficiency gains (-68.0% ACT). Crucially, on more challenging datasets (MATH-500, AIME25), DART frequently improves accuracy while reducing length. This suggests that adaptive truncation can eliminate redundant or counterproductive reasoning steps.

---

### Author Response · Authors · 2025-12-01

Dear Area Chair and Reviewers,

We sincerely thank all reviewers for their insightful and constructive feedback on our submission. In response to the reviewers’ concerns, we have conducted substantial new experiments, added thorough analyses, and clarified key aspects of our methodology. Below, we summarize the major revisions made to address the reviewers’ points:

***Key Revisions and Additions***

1. **Generalization Beyond Mathematics**
   - We added a new section **4.4: Generalization to Non-Mathematical Reasoning Tasks**, evaluating DART on three non-mathematical reasoning benchmarks—**GPQA (scientific), LogiQA (logical), and CommonsenseQA (commonsense)**.
   - **Results (Table 2)**: DART trained only on math datasets (GSM8K and MATH) achieves consistent token reduction (7.0%–49.2%) while maintaining or improving accuracy across all domains.

2. **Explicit Difficulty-Aware Analysis**
   - We conducted a fine-grained difficulty analysis using MATH-500 dataset with official difficulty ratings (Level 1–5).
   - **Results (Section 4.5.4, Figure 5)**: DART adapts reasoning length to problem difficulty—higher compression for easier problems, longer reasoning for harder ones—validating its internal difficulty-awareness.

3. **Comparison with Static Methods**
   - We reproduced **TokenSkip** and compared it with DART across all datasets and model scales.
   - **Results (Tables in rebuttal)**: DART consistently outperforms TokenSkip in both accuracy and efficiency, demonstrating the advantage of **reasoning-level adaptation** over token-level compression.

4. **Clarification of Methodology**
   - We provided detailed explanations of the distillation process in **Section 4.1** (Experimental Setup), including the full prompt templates in **Appendix B.1** (Prompt Template of Short CoT Data Distillation), and added a practical case study in **Appendix D.1** (Case Study: Short CoT Data Distillation) for transparency and reproducibility.
   - We emphasize that **model fusion is a lightweight, training-free operation**, and the final adaptive model is a single standalone model with no inference-time overhead. The entire training process for DART (including distillation and adaptive fine-tuning) requires only about 8 hours on 4 A100 GPUs, which is significantly more efficient than typical RL-based approaches that often demand dozens of GPUs and several days of training.

5. **Novelty and Contribution**
   - We emphasize DART's core innovation in addressing **difficulty-aware sufficiency** — a framework that **automatically learns the minimally necessary reasoning length for each problem**. This stands in contrast to prior static compression or uniform-length methods.
   - This capability is facilitated by our **novel data curation paradigm**, which constructs a training set where each problem is paired with its shortest correct reasoning chain. Coupled with the **theoretical foundation** of the Sigmoid length-accuracy trade-off, DART learns to dynamically allocate reasoning effort, using concise chains for simple problems and more elaborate reasoning for complex ones.
   - We contrasted DART with Prompt-based and RL-based methods, highlighting its **stability, generalizability, and model-agnostic design**.

6. **Validation of Model Fusion Effectiveness**
   - We added experiments in **Appendix C.1 (Table 5)** showing that **prompt-based methods fail to precisely control reasoning length**, unlike DART’s fusion-based approach.
   - Further validation of model fusion across different architectures is provided in **Appendix C.2**, demonstrating its consistent effectiveness regardless of the base model family.

***Additional Notes for AC’s Consideration***

*   **Reviewer DU6L’s Updated Score**: We are pleased to note that **Reviewer DU6L explicitly raised their score by two points** after reviewing our rebuttal and new experiments, indicating that their technical concerns have been adequately addressed.
*   **Addressing Reviewer dE18’s Concerns**: While Reviewer dE18 initially expressed reservations regarding novelty, computational cost, and evaluation scope, we have comprehensively addressed each point through:
    -   Cross-domain generalization experiments
    -   Explicit difficulty-aware validation
    -   Clarification of methodological details and training efficiency
    -   Clarification of novelty and contribution

    None of these issues affect the core contribution of our work, which tackles a critical pain point in the field: **enabling efficient, difficulty-aware reasoning without the instability of RL or the inflexibility of static compression**.

These revisions have significantly strengthened the paper’s contribution, clarity, and empirical support, addressing all major concerns raised by the reviewers.

We believe the revised manuscript now presents a more complete, convincing, and generalizable framework for adaptive reasoning in LLMs.

Thank you for your consideration.

Sincerely,
The Authors

---

### Meta-Review · Area_Chair_rnW9 · 2026-01-07

**Summary:**

This paper proposes DART, a supervised pipeline for difficulty-adaptive reasoning truncation. The paper targets an important efficiency problem, but initial reviews raised legitimate concerns around novelty, unclear distillation quality control, compute overhead, limited baselines, and restricted evaluation to math tasks with some accuracy drops. The rebuttal substantially improves: it adds non-math benchmarks (GPQA/LogiQA/CommonsenseQA), and direct comparisons to TokenSkip, plus clearer methodological details and a concrete compute budget. Even with these additions, the core idea still reads as a careful engineering pipeline rather than a deep methodological advance, and key claims (e.g., fusion robustness, avoidance of brittleness, and preservation of reasoning quality under truncation) remain only partially convincing without broader and cleaner evidence.

**Reviewer Concerns:**

Reviewer concerns addressed -

* Added evaluations on GPQA, LogiQA, and CommonsenseQA showing token reduction with maintained or improved accuracy.
* Added stratified results using official MATH difficulty levels (1–5) showing longer reasoning for harder problems and higher compression for easier ones.
* Reproduced TokenSkip and reported comparisons

Remaining concerns -

* Even after rebuttal, the approach largely combines known primitives (distillation + filtering shortest-correct traces + parameter interpolation) and may not clear the bar for a strong technical contribution.
* Reports indicate token reduction sometimes comes with measurable accuracy drops (notably on 7B), and the paper lacks a reliable mechanism to prevent premature truncation.

**Reviewer Scores:**

Reviewer dE18 might have increased the score by 2 points.

---

### Decision · Program_Chairs · 2026-01-26

Reject